# Using Information from Public Databases to Critically Evaluate Studies Linking the Antioxidant Enzyme Selenium-Dependent Glutathione Peroxidase 2 (GPX2) to Cancer

**R. Steven Esworthy *** and Fong-Fong Chu

Department of Cancer Genetics & Epigenetics, Beckman Research Institute of City of Hope, Duarte, CA 91010, USA; ffchu@coh.org
* Correspondence: sesworthy@coh.org

**Abstract:** Recent research on selenium-dependent glutathione peroxidase 2 (GPX2) tends to focus on possible roles in tumorigenesis. This is based on the idea that normally generated hydroperoxide species can damage DNA to produce mutations and react with protein sulfhydryl groups to perturb normal regulation of cancer-related pathways. GPX2 is one of many peroxidases available to control hydroperoxide levels. Altered *GPX2* expression levels from normal to cancer or with cancer stages seems to be the main feature in bringing it to the attention of investigators. In this commentary, we examine this premise as a basis for cancer studies, largely by trying to place *GPX2* within the larger context of antioxidant enzyme gene expression. We make use of public databases and illustrate their possible role in approaching this issue. Since use of such databases is new to us, we looked to sources in the literature to evaluate expression level data, finding general agreement with some discrepancies over the range of expression and relative expression levels among some samples. Using the database information, we critically evaluate methods used to study GPX2 in the current literature for a variety of cancers. Second, groups are now trying to compare enzymatic properties of GPX1 and GPX2 using proteins from bacterial cultures. We weigh in on these recent findings and discuss the impact on the relative GPX2 and GPX1 functions.

**Keywords:** GPX2; GPX1; PRDX; catalase; cancer; tumor-derived cell lines; public databases

## 1. Introduction

### 1.1. GPX2 in the Scheme of Things

Claims for selenium-dependent glutathione peroxidase 2 (GPX2) involvement in tumorigenesis constitute about one third of all GPX2 publications (PubMed). GPX2 could play a role in tumorigenesis based on older and somewhat stronger work that considered the selenium (Se) dependance of the enzyme for in vitro studies and maintained some awareness of competition for hydroperoxide (ROOH-hydrogen peroxide and alkyl hydroperoxides) metabolism among antioxidant enzymes (minimally GPX1, 2 and 4) [1–9]. In our view, the current work is often weak, repeatedly making the same errors. It is because of this situation that we want to present a commentary on the current work and suggest easy ways to improve studies, some of this drawing on the use of public databases [10].

As a selenium-dependent glutathione peroxidase, GPX2 efficiently reduces hydroperoxides at the expense of glutathione (GSH) in the standard coupled enzyme activity assay, showing the possibility that it helps control steady-state levels of ROOH in cells (our conditions were -pH 7.3, 3 mM GSH, 60 μM hydrogen peroxide, Na Azide added to kill catalase (CAT); coupled assay-glutathione reductase and NADPH) [11]. We found that in mice lacking *GPX1* and *GPX2* (GPX1/2-knockout mice mutation levels in the ileum and colon mucosal epithelium were significantly elevated; one factor in promoting often cancerous tumors found in the ileum and colon [12]. This is presumed to be due to lack of control over levels of hydroperoxides. In fact, knocking out major oxidant generating sources

in the intestine, NADPH oxidases NOX1 or DUOX2, lowered the incidence of ileocolitis in GPX1/2-DKO mice [13,14]. ROOH species are known to oxidize sulfhydryl groups of enzymes affecting activities that can impact tumorigenesis [15]. The prostaglandin pathway, known to stimulate growth of colon tumors, may also be impacted by expression levels of GPX2 [6]. However, for GPX2 to have an effect it must be highly expressed relative to competing peroxidases, including its sister isoenzyme, GPX1. Increases or decreases in expression levels are often substituted for this condition as the premise behind many investigations into the relationship between *GPX2* and cancer. Such work is often supported by misuse of the DCFH-DA assay (2′,7′-dichlorodihydrofluorescein diacetate) in cell lines after forced manipulation of levels by si- or sh-RNAs or overexpression and utilizing culture media generally lacking in selenium (Se), which detrimentally impacts GPX2 and GPX1 at the activity level [16–21].

In July of 2022, we published a review of progress in GPX2 studies over the 30 years since its discovery [21]. In the critical appraisal section, we pointed out that *GPX2* has a limited expression range and is always in the company of *GPX1*, peroxiredoxins *1–6* (PRDX-thioredoxin-dependent peroxidases) and catalase (CAT), which will compete for a reduction in ROOH based largely on relative protein abundance and compartmentalization (constraint for CAT) [22]. We suggested that the relative contribution of GPX2 to ROOH metabolism could be evaluated before studies began. This would serve to apprise the likelihood that GPX2 could be a major factor in ROOH catabolism and whether any changes in expression levels were likely to be impactful. This could be estimated using such databases such as The Human Protein Atlas (THPA), The Cancer Gene Atlas (TCGA), Tumor Immune Estimation Resource (TIMER 2.0) and Gene Expression Profiling Interactive Analysis (GEPIA2) and for human tumor-derived cell lines using the Cancer Dependency Map Project (DepMap). These sites supply expression levels as TPM or FPKM and we will examine these metrics as surrogates for protein or activity levels and how see how they compare to measurement of mRNA levels via RT-PCR in several publications.

### 1.2. Early Characterization and Current Updates on GPX2

A major point is *GPX2* was not found by pursuing any hypothesis or trying to rectify any data anomaly. It came as result of an effort to clone human *GPX1* from a human liver cDNA library [21]. We found that the cDNA clone, when transfected into MCF-7 cells encoded an active, cellular selenoperoxidase with a hydroperoxide substrate range like GPX1. The physical properties were only slightly different from GPX1 [23]. We inferred that any presence of GPX2 in tissues would be obscured by co-expression with the ubiquitous GPX1. The human *GPX2* gene has not acquired mutations likely to impair function and the common polymorphisms do not seem to have the reputed impact of pro → leu 198 (now 200) of *GPX1* in cancer, which while found in the *GPX2* gene is rare (4.3%, heterozygotes only) [24–28]. This leaves perceived changes in expression levels from normal to cancer or with cancer stages as the main factor in bringing *GPX2* to the attention of many investigators [21].

Schwarz et al. reported that recombinant GPX2, expressed in bacteria, had 10-fold less activity than GPX1 and extended our findings on substrates, showing a nearly identical substrate range for a large spectrum of ROOH species [29]. We would presently dispute the low value reported by Schwarz et al. based on our estimate that the specific activity was 3-fold lower [30]. As indicated in an analysis presented in a section below, using the 10-fold-lower specific activity difference is as adequate a fit to cell line expression levels from the DepMap project (https://depmap.org/ (accessed on 1 July 2023)) as the 3-fold-lower estimate. The coupled assay used by us and Schwarz et al. is limited by the apparent Km for GSH and not by the peroxide substrate, which is in vast excess (>50 μM) [11]. The case for possible lower specific activity of GPX2 is the substitution of Arg 185 (Bovine GPX1 AA reference) with threonine and Lys 91 (Lys 91B-other subunit) with Gln possibly affecting GSH binding [31,32]. This property of GPX2 is unlikely to impact its performance, in vivo, relative to GPX1 as GSH levels in cells should suffice to keep GPX2 fully reduced [33]. The

forward rection between Se-cysteine and ROOH is diffusion limited and unlikely to vary between GPX1 and GPX2. In cells at the normal physiological GSH concentration range (1–5 mM), ROOH concentrations are limiting (<0.1 μM) [11,22].

A second update to the GPX2 story is the identification of a possible protein–protein interaction between GPX2 and PCBP2, which is proposed to stabilize GPX2. This is the first such interaction reported [34].

### 1.3. GPX2 Is Unlikely to Be Unique in Its Action

We will be commenting about papers where *GPX2* levels were manipulated for experimental purposes. As the presumed effectors of any biochemical action of GPX2 are ROOH, the manipulation of *PRDX1-6* might achieve the same goal. All indications are that GPX1 and GPX2 should be largely interchangeable in such experiments and one recent paper demonstrated this [34]. An examination of the literature shows studies where cellular resistance in MCF-7 and prostate tumor-derived cell lines to $H_2O_2$, ROOH and doxorubicin was impacted by GPX1, PRDXs 1-6 and CAT [35–40]. Prostate-derived cell lines exposed to $H_2O_2$ and tert-butyl hydroperoxide showed reduced viability after si-RNA silencing of *PRDX1*, *PRDX2* or *PRDX3*, but not *PRDX4*, as part of a single study. Suppressing GSH-dependent pathways by lowering GSH levels with BSO produced a similar increase in susceptibility to $H_2O_2$ and tert-butyl hydroperoxide [40]. Similarly, in the azoxymethane-DSS rodent model of colon cancer, five studies showed *GPX1* (+*CAT*), *PRDX1* (silencing), *PRDX2*, *PRDX4* and *PRDX6* gene knockouts (KOs) had fewer tumors, an effect ascribed to increased expression of regulatory T cells (where *GPX2* is not expressed) and suppressed inflammatory responses [41]. The idea of access to common ROOH pools with observed shared outcomes after manipulation of expression levels can be inferred from additional sets of studies involving cell lines, animals and data from human subjects [42–53]. The key factors would be relative protein levels as the peroxide rate constants are comparable [54]. Alternative actions for *GPX2* might include preventing PRDX inactivation (possible at low levels of expression), the mRNA acting as bait for non-coding RNA species, direct protein–protein interactions, or an impact on expression of other proteins by still unknown mechanisms (p53 and MMP-9, suggested as targets in reference [55]) [21,34,55–59].

### 1.4. Compartments and Conditions for Major Impact by GPX2

In our 2022 review, we examined reports on basal cells (including stem cells or proliferative compartments) where GPX2 protein abundance might rival that of other antioxidant enzymes [21,60]. We suggest, since *GPX2* expression is occurring as part of packages (e.g., stem cells, Paneth cells, cholangiocytes, certain cancers) with different compositions of co-expressed genes and down-stream ROOH sensitive protein targets, GPX2 might appear unique in its impact even in the presence of GPX1 and PRDXs. We find it doubtful that GPX2 would have any unique functions in normal or cancerous cells simply based on one of the first major observations made about *Gpx2-*, this being that to induce ileocolitis in mice with high penetrance, *Gpx1* had to be knocked out as well [61–63]. That PRDXs and catalase failed to offer protection in this case seems to be due to a major shift in relative antioxidant enzyme expression in basal cell compartments in favor of GPX2 as opposed to PRDXs [60]. In most papers dealing with antioxidant enzymes and tumorigenesis, the full context of antioxidant enzyme expression fails to be explored leading to the blind men and the elephant situation where multiple papers implicitly suggest that one antioxidant enzyme or another is the key factor in similar states of cellular stress [21].

## 2. Application of Public Database Information in Studies of GPX2

### 2.1. Pre-Study Evaluation of Antioxidant Enzymes in Normal Tissues, Cancers, and Cancer-Derived Cell Lines

Almost everyone has been made aware of the many publicly accessible databases where expression levels of genes are compiled for normal tissues, cancers, and cancer-derived cell lines (some examples-THPA, TCGA (https://www.cancer.gov/ccg/research/

genome-sequencing/tcga (accessed on 1 July 2023)), TIMER 2.0 (http://timer.cistrome.org/ (accessed on 1 July 2023)), GEPIA (http://gepia.cancer-pku.cn/ (accessed on 1 July 2023)), Cancer dependency map project (https://depmap.org/portal/ (accessed on 1 July 2023)) and the CCLE (https://sites.broadinstitute.org/ccle (accessed on 1 July 2023)). With limitations that will be noted, these sites can be used to compile expression profiles for antioxidant genes without the tedium of in-lab profiling with some uncertainty which might be resolved with IHC or tumor purity analysis [64]. The goal is to create a crude, yet effective consensus profile of antioxidant enzyme gene expression as relative TPM that can address the following question: can GPX2 contribute in a substantial way to ROOH metabolism by virtue of significant levels of expression in specific normal tissues and cancers?

To have perspective for the remainder of the discussion, we note that total antioxidant enzyme gene TPM (*GPX1*, *GPX2*, *PRDX1-6* and *CAT*) in GEPIA ranges from 1000 to 3000 TPM (also TIMER 2.0; mean total TPM 2095 $\pm$ 565 std). The totals differ from site to site. THPA/TCGA has a lesser total based on using FPKM, ranging from 598 to 2000 (rough conversion factor of FPKM $\times$ 2 = ~TPM); however, the relative values tend to remain constant. The choice of GEPIA/TIMER 2.0 is due to the agreement between the sites for tumor values and DepMap for cell lines, all in TPM. Our portrayals of tumor expression are based on the raw median TPM values with no accounting for proportions of tumor vs. non-tumor in samples. We note that in examining the tumor purity data set from TCGA (https://gdc.cancer.gov/about-data/publications/pancan-aneuploidy (accessed on 1 July 2023); https://api.gdc.cancer.gov/data/4f277128-f793-4354-a13d-30cc7 fe9f6b5 (accessed on 1 July 2023)), the range for samples in the first 850 tumors listed was 0.13–1 (mean 0.63 $\pm$ 0.20 std). The justification for this unfiltered type of portrayal is that antioxidant enzymes genes are often highly co-expressed with *GPX2* in cancer epithelium, although *GPX1* and *PRDX1-6* and *CAT* are also expressed at lower levels outside the tumor boundaries. We will generally underestimate the potential contribution of *GPX2* to the tumor total antioxidant enzyme gene TPM. For a consensus tumor-derived cell line profile from the DepMap project, the total TPM was 2045 using median data. As to defining a significant TPM count for *GPX2*, we would arbitrarily set a threshold of 64 TPM (32 FPKM) as the lower end, generally accounting for 3–6% of total TPM, depending on the cancer or cell line. We have portrayed the profiles as each antioxidant enzyme gene relative to the total TPM, rounded to 5% increments (Figure 1E).

### 2.2. Strengths and Limitations of Available Database Information

We will look at results compiled from the different databases for normal tissues and cancers in comparison to each other and for tumor-derived cell lines in comparison with tumors and RT-PCR and activity results from publications. This information will be used to evaluate the limitations of the databases as sources of information for a first approximation of antioxidant enzyme contributions to ROOH metabolism.

Figure 1 illustrates the limited tissue expression range of *GPX2* versus *GPX1* and *PRDX1* (median TPM metric; TCGA datasets from THPA), generally confirming early impressions from Northern analysis that *GPX2* expression is quite limited [23].

Note that for normal tissues, there is some disagreement over the median TPM/FPKM metrics among the available database sites (Figure 2). GEPIA values for colon and rectum are exceptionally low, while the TCGA level for liver is high. Finally, TIMER2.0 has a much higher value for pancreas than the other two sites. Correlations yield an R-square value of 0.35 for GEPIA vs. TCGA, 0.47 for TCGA vs. TIMER 2.0 and an abysmal 0.067 for TIMER 2.0 vs. GEPIA in the correlations of the 12 solid tissues sources of the most common cancer types based on the NCI cancer statistics page (https://www.cancer.gov/about-cancer/understanding/statistics (accessed on 1 July 2023)). Both GEPIA and THPA (TCGA) include many normal colon and rectum samples with nearly zero levels of *GPX2*. GEPIA may lump sigmoid colon data (35 TPM; THPA value) with transverse colon data (258 TPM). For rectum, TIMER2.0 and THPA (TCGA) seem to have only a handful of

normal samples, while GEPIA shows 349 rectum samples. Variation in sample numbers and bias in small sample number sets may explain some of the issues, for example, four pancreas samples in TIMER2.0. As to how samples can have very low values, we note that for normal bladder and colon, one IHC sample each in THPA barely contains any epithelium and the range of potentially *GPX2* expressing glandular cells in the small set of examples of normal colon histology (H&E) displayed in THPA is 5–65%. The agreement of the sites for median TPM/FPKM for the 12 most common solid cancers is excellent, with a uniform R-square value of 0.99 among the comparisons (Figure 3A–C).

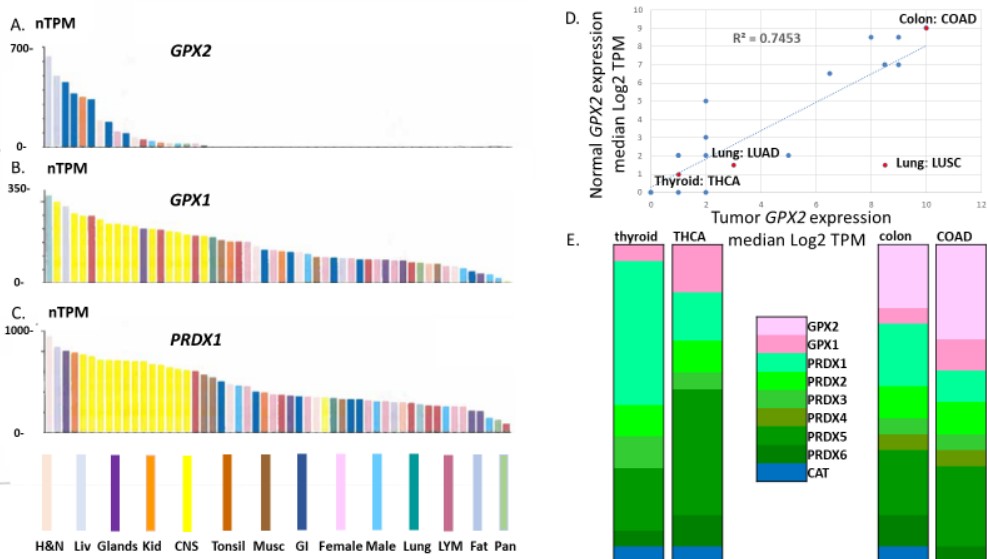

**Figure 1.** Limited *GPX2* expression across normal tissues compared to *GPX1* and *PRDX1*. Panels (**A**–**C**) TCGA data set extracted from THPA and organized by highest to lowest expression levels. Panel (**D**) shows the correlation of normal tissue expression levels and derived cancers as log2 TPM of the median expression levels. Panel (**E**) shows the relative antioxidant enzyme gene expression levels (*GPX2*, *GPX1*, *PRDX1-6* and *CAT*) compared to total median TPM for normal thyroid and cancer (THCA) and normal colon and cancer (COAD). These have not been adjusted for proportion of *GPX2* expressing cells.

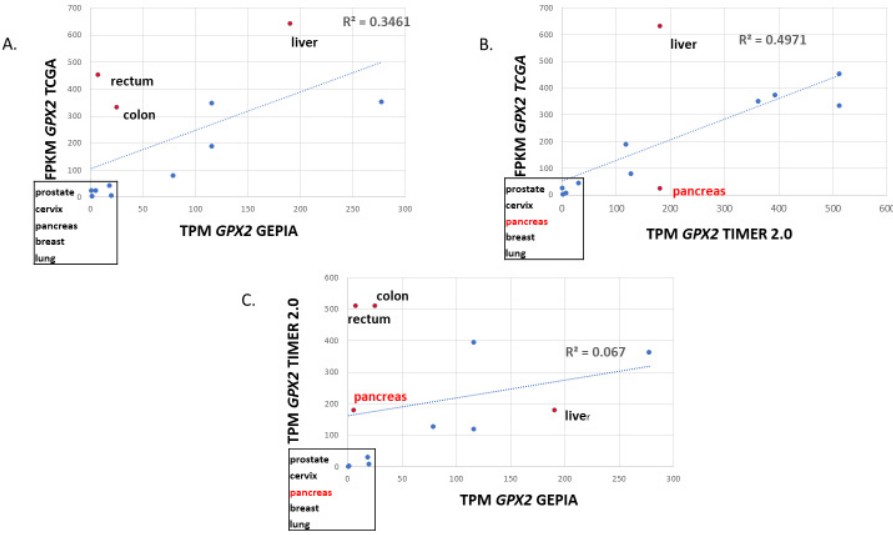

**Figure 2.** Correlation of the median TPM or FPKM metric for *GPX2* expression in normal tissue sources of the 12 most common cancers for TCGA vs. GEPIA, TCGA vs. TIMER2.0 and TIMER2.0 vs. GEPIA (panels (**A**–**C**)). Box highlights low-expressing tissues, with RED indicating the exception of pancreas in TIMER2.0. In the graphs, red dots show the major discrepancies among the databases.

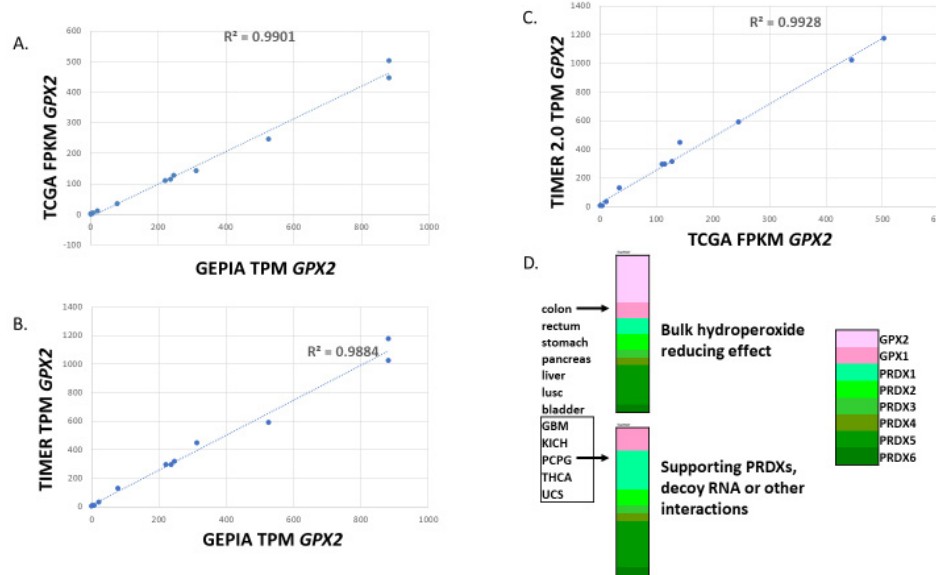

**Figure 3.** Correlation of the median TPM or FPKM metric for *GPX2* expression in the 12 most common cancers for TCGA vs. GEPIA, TCGA vs. TIMER2.0 and TIMER2.0 vs. GEPIA (panels (**A–C**)). Panel (**D**) shows examples of median TPM for each antioxidant enzyme gene relative to total TPM for cancers in the categories of high *GPX2* expression and low expression. Based on median values for *GPX2*, 2 categories of mechanism can be proposed, functioning in a major ROOH catabolism role at high levels of expression or at low levels of expression preventing PRDX inactivation, tying up non-coding RNAs or protein–protein interactions. Of course, high expression does not preclude other functions.

*GPX2* expression has enormous variation in tumors when compared to the other antioxidant enzyme genes. For the latter, the standard range of TPM/FPKM values is 2–16-fold (occasionally 32-fold), while for *GPX2*, the range is 2–4000-fold (Figure 4).

This means that for many cancer types, low median values do not preclude a fair number of high-expressing tumors and vice versa. Based on variation in tumor purity being limited to 8-fold across samples, it seems likely that there is genuine variation in cellular *GPX2* expression levels among tumors within each group. There is some correlation between the median *GPX2* levels in normal tissues (log2 TPM) and the resulting cancers (R-square, 0.74; Figure 1D). This largely reflects limitations in the possibility *GPX2* can be highly expressed for tumors from tissues like brain (glioma), thyroid (see profile; Figure 1E), kidney, and thymus as opposed to colon (profiled in Figure 1E). Lung/lung squamous cell carcinoma (Lung: LUSC; Figure 1D) is a notable exception. This constraint generally applies to tumor-derived cell lines from low-expressing tissues; glioblastoma/Brain/CNS with one line = 22 TPM, most < 2TPM; kidney with one line at 155 TPM, the remaining 31 < 1 TPM; lymphoid and myeloid lines < 2 TPM; thyroid < 2 TPM (Figure 4A; DepMap). Also, a common trope for cancer studies is the comparison of normal to cancer expression levels. This presents two problems with *GPX2*. First, there are a few cancer types where the median expression level is approximately the same as the normal tissue level. Yet, the range of expression levels is so large in the tumor sets that it makes sense to pursue a role for *GPX2* in tumorigenesis based on the division of low vs. high expression (Figure 4A). Second, the idea of the normal tissue levels as comparable to tumor levels is precluded by the limited range of localization of *GPX2* in many normal tissues and the issue of which cells are the origin of the tumors. This may be seen in the liver where there are non-expressing hepatocytes and moderately expressing cholangiocytes (THPA; IHC: higher expression in bile duct and gall bladder). Derived tumors are found from both origins spanning the range from no expression to over 500 TPM (sizable level of TPM as a percentage of total antioxidant enzyme TPM) representing no change to huge up-regulation in hepatocarcinoma and lower

expression, no changes or up-regulation in cholangiocarcinoma (insert, Figure 4A). The problem is more likely to be an issue in the cases of breast, lung, and prostate cancer, where *GPX2* expression is normally more confined, and the source of the tumor cells may be obscure. In most tissues, GPX2 is confined to sub-tissue sites, commonly the epithelium of the tissues (urinary bladder, Figure 5) with further restriction to the base of the crypts and glands of the small intestine and colon epithelium, basal cells in breast, lung and prostate [21].

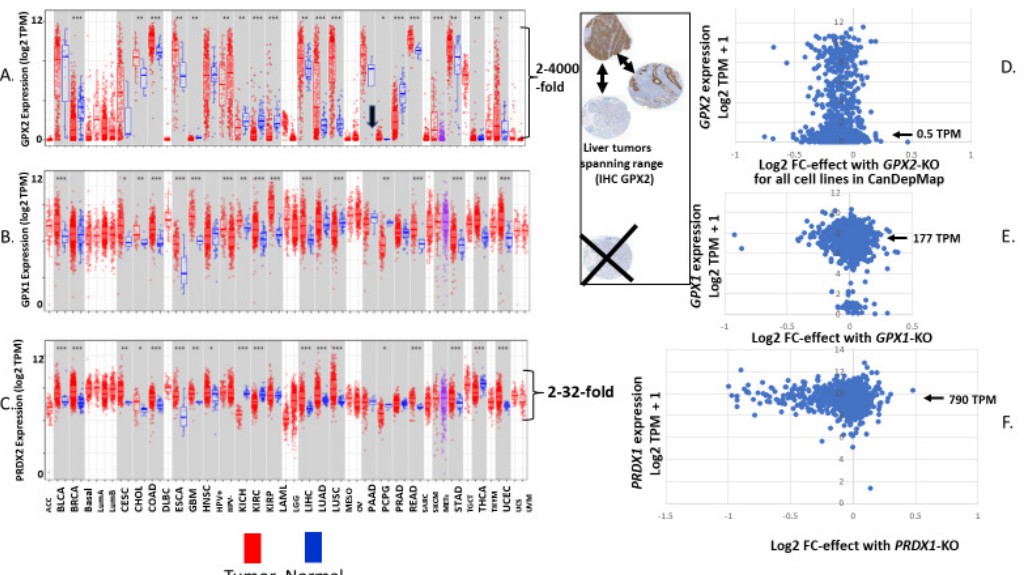

**Figure 4.** Panels (**A**–**C**) show expression of *GPX2* (**A**), *GPX1* (**B**) and *PRDX1* (**C**) in samples of normal and derived tumors as log2 TPM. Figure retrieved from TIMER2.0. Arrow in panel (**A**) highlights the discrepant high normal pancreas levels specific to TIMER2.0. Panels (**D**–**F**) are expression level vs. effect plots generated for tumor-derived cell lines using data available at DepMap; *GPX2* (**D**), *GPX1* (**E**) and *PRDX1* (**F**). Expression levels are log2 TPM + 1 and the effect is log2 fold change (FC) meaning the impact of CRISPER-CAS9 knockout of the gene on the ability of cells to replicate or survive. The arrows in panels (**D**–**F**) show the median values for all cell lines as TPM for each gene. Comparable sets of graphs for the remaining antioxidant enzyme genes are found in figures that follow. The insert shows that two bases for variation in the LIHC set is due to the proportion of stained cells (tumor purity) and the wide range of staining intensity. The range found for *GPX2* is extreme, while for the other enzymes, it is much less, precluding absence of expression (Xed out IHC panel). Tumor only sets and less common tumors are in smaller type.

Colon presents an interesting example of the problem of comparing normal to tumor. LGR5 and/or ASCL2 co-expressing cells at the gland base of the colon are the source of 75–85% of colon tumors [65]. This is also the site of GPX2 expression based on IHC [21,65]. From the available IHC, we would very roughly estimate the range of the high-expression zone of GPX2 to be 5–10% of the total tissue by volume with a very limited variation, while in tumors the range would be 13% at the low end with a median value of 62%. Normal colon has a median *GPX2* TPM of 422 × (10–20) = 4022–8044, extrapolated to a purity of 1 (excluding the GEPIA estimate). Cancerous colon has a median value of 958 × 1.6 = 1533 TPM, suggesting no up-regulation from normal to cancer, even including values of 2900–4000 TPM at the highest range of expression, which we would expect to represent tumor purity values closer to 1. However, we have the outlier GEPIA value for normal colon, whereby the extrapolated values would be 250–500 TPM, suggesting that up-regulation would be occurring in a large fraction of colon cancers. In any event, the likelihood that *GPX2* expressing colon cancer occupies 6 to 12 times more tissue volume than in the normal case does require consideration in estimates of up-regulation. This ignores the 15–25% of tumors originating outside of the LGR5 and ASCL2 expression zone,

for which up-regulation of *GPX2* expression might be occurring. This division of colon cancer sources has not been addressed in GPX2 studies.

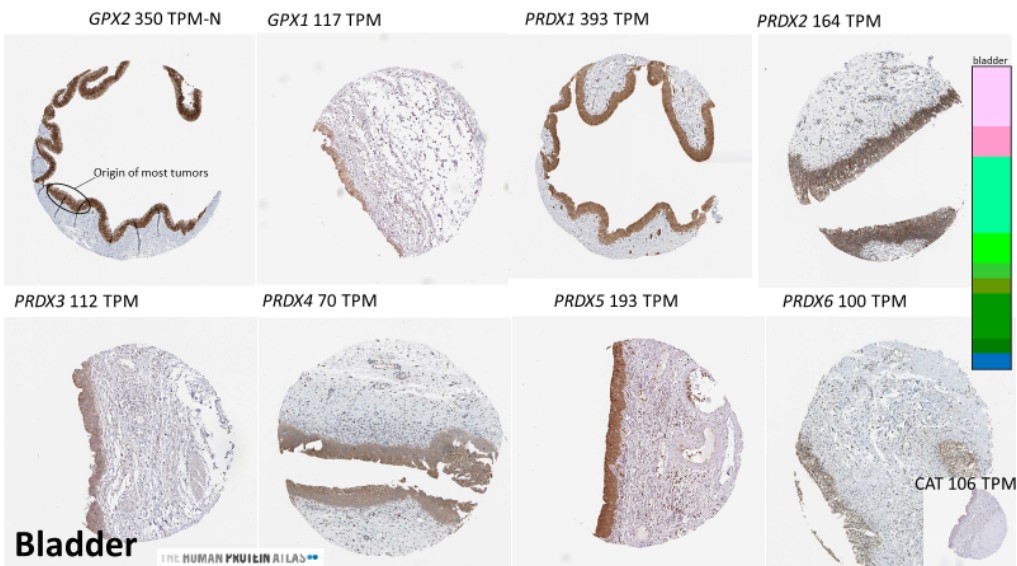

**Figure 5.** Examples of the expression sites of antioxidant enzyme proteins using IHC in normal bladder. Images taken from THPA. The figure illustrates epithelium as the locale of high expression for the enzymes and shows that while GPX2 is limited to this site, the other enzymes show low-level expression outside of the epithelium. Brown indicates GPX2 localization, hematoxylin for tissue.

### 2.3. Building Antioxidant Enzyme Expression Profiles for Tissues and Derived Tumors

Some of the points raised above pertain to the issue of compiling expression of all the antioxidant enzyme gene expression levels to build profiles so that GPX2′s contribution to ROOH metabolism can be assessed. Assessments in some normal tissues and tumors can be made, since IHC suggests similar sub-tissue localization for high-expression sites across the enzymes as illustrated in bladder for normal tissue and colon for tumor tissues (Figures 5–8). We must emphasize the value of examining the publicly available IHC for antioxidant enzymes when studying their role in cancer.

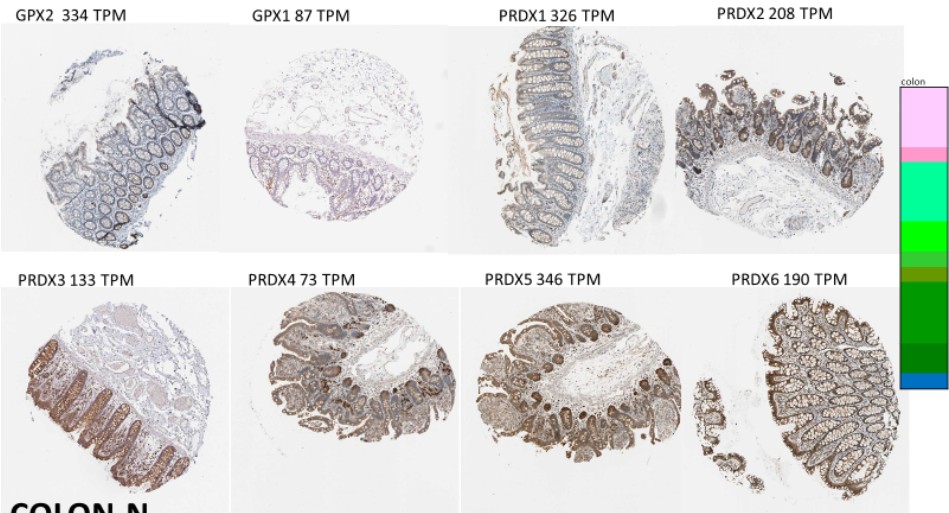

**Figure 6.** Images of normal colon, illustrating IHC for the antioxidant enzymes. The enzymes tend to share high expression in the epithelium (better images of normal colon for GPX2 IHC can be found in references in [21]). However, GPX1 and the PRDX expression spills out into the submucosa. IHC did not detect CAT. See Figure 5 for image colors and interpretation.

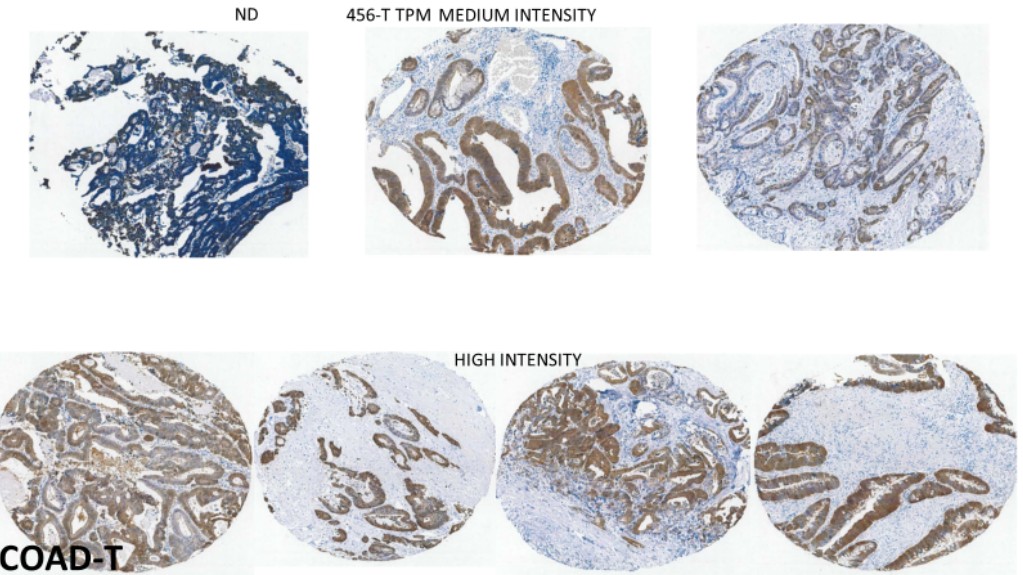

**Figure 7.** The localization of the GPX2 sites of high expression by IHC tend to be preserved in COAD as in colon (source: THPA). The one example of non-expression (ND) looks to be a badly prepared sample. The remaining examples reflect the high median TPM, and lesser variation shown in Figure 4A for COAD. This set shows a little less range in tumor proportion than some other cancers with variation found in the general intensity from sample to sample and variation within the sample fields. See Figure 5 for image colors and interpretation.

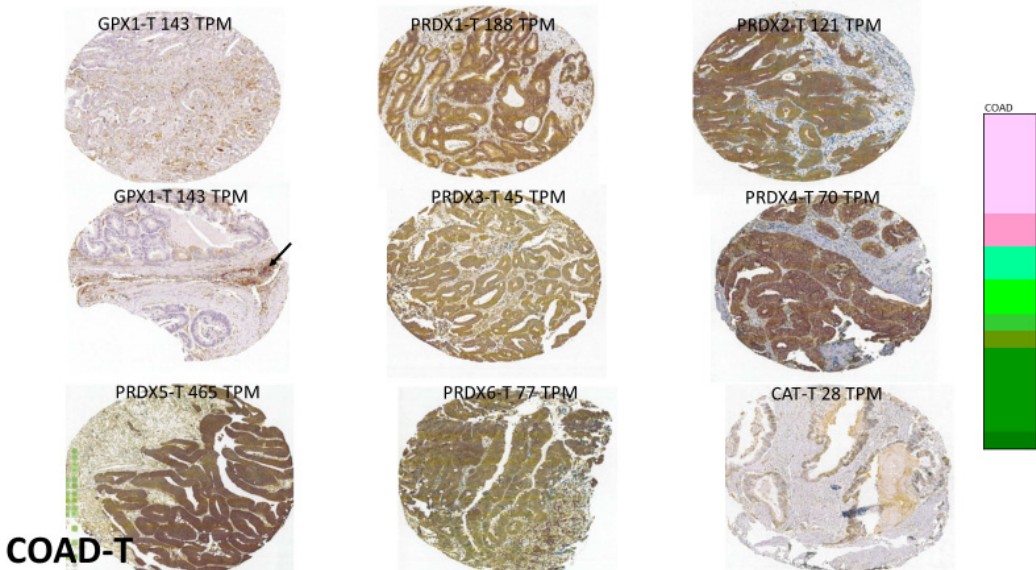

**Figure 8.** Samples of GPX1 and PRDX expression sites in COAD (source: THPA). These images were selected to highlight the tendency for high-expression sites to mimic that of GPX2, while again sometimes straying outside of the epithelium-like compartment (arrow). See Figure 5 for interpretation.

### 2.3.1. Tissues with Clearer Zones of GPX2 Expression and High Expression Levels

Epithelial compartments of bladder, colon and stomach are a common site of high expression among the antioxidant enzyme set, although *GPX1* and the PRDXs are expressed outside of epithelial cells compartments so that some adjustments need to be made to rectify estimates for *GPX2* as a fraction of the total. For tumors from the bladder and stomach, high expression of most of the antioxidant enzymes occurs in the epithelium, and a similar adjustment could be made in the compiled profiles to account for expression

outside of the epithelial compartments based on available IHC at THPA, for example. Adjustments for normal colon and liver would be more difficult given the restricted range of *GPX2* expression within the tissues, while the other enzymes show more even expression throughout the epithelium of colon and in hepatocytes (THPA) [63]. In the cancer tissues, antioxidant enzyme gene profiles would have more value with the somewhat more consistent expression of GPX2 throughout the samples.

### 2.3.2. Tissues with Low and Less Certain Range of GPX2 Expression

Breast, lung and prostate present difficult instances where *GPX2* (and GPX2 protein) can barely be detected, or there are discrepancies for the site of expression between publication and database IHC [60,66,67]. Some of this originates based on choices of antibodies in THPA. On occasion, results from 2 or 3 different antibodies are shown side by side with marked differences in ability to detect GPX2. Despite the lack of clear localization of GPX2 in breast via IHC (THPA), one paper used cell fractionation of luminal and basal compartments to localize GPX2 in the basal compartment (median tissue TPM < 16) [60]. It is possible to use the data in that paper to build the antioxidant enzyme profile for the breast basal and luminal compartments as protein abundance [21,60]. This exercise reveals GPX2 to be highly represented relative to GPX1 and PRDXs in the basal compartment (see graphic abstract in ref. [21]; left-hand panel based on the data from reference [60]). For prostate, IHC at THPA does not detect GPX2 (same antibody used for breast samples and lung cancer; see below). However, one publication seems to show uniformly strong GPX2 staining in the basal cell compartment, which is more consistent with the median tissue TPM of ~32 [66]. Unlike breast tissues, we have no alternative resource to aid in building profiles of the antioxidant genes or protein levels [60]. However, use of THPA IHC images greatly assists as in the example of bladder shown in Figure 5. Generally, the antioxidant enzymes are highly co-expressed in the glandular compartment, which includes the basal cells. There is some expression outside of the glandular cells for which some adjustment must be made. This still leaves the issue of which cells express *GPX2* in normal prostate with NKX3-1 positive cells embedded in the glandular compartment being candidates [68]. This marker does not seem to predict *GPX2* expression in prostate tumors (THPA). *TP63* expression seems to be a closer match to normal vs. tumor expression for both breast and prostate with levels tending to decline from normal to tumor [60,69]. In prostate and breast, the net impact in cancer appears to be down-regulation of *GPX2* median expression to 6 TPM or less. There are breast tumor samples with expression levels up to and more than 256 TPM and rarer prostate samples with values up to 64 TPM. Adjustments for tumor purity would probably place a few more samples from both sets in the significant expression range. However, the likelihood of marginal expression levels requires more refined investigation for the tumors from these tissues covering very low levels of expression and addressing the question of the sources of the tumors. In the breast, basal cells may be responsible for only 15–25% of all tumors, although they are aggressive [70]. It is not clear that all basal-cell like breast cancers derive from the stem cell population. If so, it appears that such cells must lose *GPX2* expression as the cancer progresses as they also lose TP63 expression [71]. Among breast cancer-derived cell lines, it is in the *TP63* minus (ER+/−; PGR-; HER2+/−) set that *GPX2* expression is highest, although there is a slight positive trend for a small set of the lines (DepMap). Prostate cancer-derived cell lines show a stronger association between *TP63* levels and *GPX2* expression (DepMap). It should be possible to build somewhat meaningful antioxidant enzyme gene profiles for breast and prostate tumors due to more uniform GPX2 expression across the tumor tissue (IHC, THPA).

Normal lung presents one more case where the antibody used in THPA did not detect GPX2. After induction via the stress of allergic inflammation in mice, which induced *Gpx2* levels 12-fold, GPX2 protein (IHC) was detected in the basal cells of lung, but not before (median human lung tissue TPM < 4) [67]. In one study of non-small cell lung cancer (NSCLC; LUAD and LUSC), the authors noted that GPX2 was detected by IHC in only

14 of 252 normal samples and in those only "sporadically in bronchiolar epithelial cells", consistent with the THPA results [72]. Like prostate, we lack the tissue fractionation data available for breast and would have a difficult time constructing a reliable antioxidant enzyme profile for normal tissue. In fact, lung appears to be the most difficult case due to the presence of high abundances of antioxidant enzyme expressing macrophages (no GPX2 expression) and the presence of a mixed population of alveolar cells with variable expression profiles (IHC, THPA). For lung cancers, the IHC at THPA does show significant staining for both LUAD and LUSC samples (the same antibody described above). There is a huge ambiguity in assessing *GPX2* in NSCLC. There is one instance where the *GPX2* expression is highly likely to be consistently increased at the cellular level in cancer. This would be lung squamous cell carcinoma (LUSC 128-fold; normal median expression 2.8 TPM vs. tumor 362 TPM) (Figure 1D). Lung presents an interesting instance where the decision to pair normal to tumor samples may have led to a correct inference about generally elevated *GPX2* levels in lung squamous cell carcinoma tumors (two exceptions in paired sets; see Figure 1F and Figure 4A in [72]) [72]. The parallel inference for lung adenocarcinoma (LUAD) is a little less sound with a fair fraction of the pairs showing lower levels in the tumors (their Figure 1E; two-fold-higher median expression level in tumors vs. normal)**.** On the one hand, we note that the gap between the highest expressing normal lung sample in TIMER 2.0 and highest expressing LUAD samples is in the range of 32- to 64-fold, and that in mouse lung GPX2 could not be detected without stress to the lung tissues [67]. On the other hand, the median *GPX2* expression level in the LUAD group is less than 16 TPM, which is below the upper range of normal samples. This is generally unlikely to yield significant expression even accounting for low tumor purity (see Section 2.7). In squamous cell carcinomas, the median *GPX2* expression level is nearly equal to *GPX1*. Accounting for *GPX1* and *PRDX1-6* expression outside of the tumors, GPX2 would appear be a significant part of ROOH metabolism in LUSC, while in LUAD, it is unclear. The difference in the adenocarcinoma and squamous cell carcinoma outcomes may relate to different cell types producing the tumors [73]. Here again, one of the points in the apparent elevation of expression in tumors is the spread of expressing tumor cells into regions of the lung barely expressing *GPX2*.

### 2.3.3. Esophagus Has Moderate Normal Levels GPX2 Expression Levels

Esophagus tissue has a median expression level of 100 TPM. As in normal breast, lung and prostate, the sites of expression are quite confined, although GPX2 was just detectable with a working antibody [74]. Cancer samples generally showed greater staining intensity than distantly sampled normal tissue, although not proximally sampled tissue. The staining intensity in IHC agrees with the cancer TPM data showing an extreme range of expression is possible with a high median value of 500 TPM and detectable staining in all but a few samples. Building a profile for normal appears to be like that for bladder, breast and prostate, with high antioxidant enzyme expression confined here to the squamous epithelium in which the *GPX2* expressing cells are imbedded (IHC, THPA). Building a useful cancer profile seems feasible for esophageal cancer.

### 2.4. Selenium as a Variable in Cancer Studies and Use of Cell Lines

If subjects of cancer studies are from the USA, Canada and portions of Central and South America the Se status will be rated as sufficient as is also the case for India and some portions of Europe [75]. However, parts of Europe and Asia might have subjects with low Se status (1 in 7 worldwide) [76]. *GPX1* mRNA is subject to nonsense-mediated decay when Se levels are low [77]. The issue does involve both GPX1 and GPX2 protein and activity levels. This is more problematic for *GPX2*, since the mRNA levels are presumably less likely to be diminished by low Se levels while the protein and activity would be impacted, creating additional discrepancies between the TPM metric and the likely contribution of GPX2 to ROOH metabolism [77–79]. To partially address this latter point, we also supply

*GPX1* expression levels as a more direct basis of comparison of GPX2 contribution to ROOH metabolism for cell lines we have used or are reported on in GPX2 papers (Table 1).

**Table 1.** *GPX2* and *GPX1* TPM and fraction total antioxidant gene TPM for cell lines.

| Cancer | Cell Line | THPA GPX2 TPM | DepMap GPX2 TPM | GPX2 Frac. | GPX1 | GPX1 Frac. | Cancer | Cell Line | THPA GPX2 TPM | DepMap GPX2 TPM | GPX2 Frac. | GPX1 | GPX1 Frac. |
|---|---|---|---|---|---|---|---|---|---|---|---|---|---|
| **bladder** | bcb3 | 1543 | 1330 | 0.24 | 218 | 0.039 | **lung** | | | | | | |
| | vmcb1 | 68 | 83 | 0.039 | 181 | 0.084 | | a549 | 767 | 656 | 0.15 | 100 | 0.02 |
| | kmbc2 | 677 | 782 | 0.29 | 156 | 0.058 | | ncih1385 | 1385 | 1236 | 0.45 | 55 | 0.02 |
| | umuc5 | | 29 | 0.014 | 99 | 0.048 | | ncih1573 | 888 | 1053 | 0.2 | 89 | 0.017 |
| | umuc14 | 782 | | 0.29 | 347 | 0.115 | | ncih358 | 3.1 | 4.37 | 0.002 | 299 | 0.13 |
| | | | | | | | | sw1573 | 1.4 | 1.73 | 0.0007 | 331 | 0.15 |
| **breast** | skbr3 | 32 | 16.6 | 0.006 | 0.64 | 0.00012 | | ncih2291 | 0.6 | 1 | 0.00004 | 872 | 0.34 |
| | mcf7 | 4.5 | 0.97 | 0.00036 | 0.57 | 0.0002 | | ncih1792 | 0.1 | 0.36 | 0.0001 | 255 | 0.075 |
| | hs578t | 0.2 | 2.5 | 0.008 | 217 | 0.086 | | ncih23 | 0.2 | 0.25 | 0.00002 | 0.86 | 0.0006 |
| | zr75-1 | 3.4 | 4.44 | 0.0006 | 8.23 | 0.0016 | | ncih520 | 359 | 390 | 0.055 | 62 | 0.009 |
| | mdamb175 | 703 | 1125 | 0.3 | 0.65 | 0.0002 | | ncih460 | 0.5 | 0.75 | 0.0004 | 256 | 0.14 |
| | du4475 | 320 | 363 | 0.11 | 342 | 0.117 | | pc9 | 309 | | 0.094 | 349 | 0.14 |
| | mdamb231 | 0.9 | 1.6 | 0.0003 | 381 | 0.132 | **prostate** | | | | | | |
| | mdamb157 | 0.6 | 0.75 | 0.00025 | 250 | 0.104 | | Du145 | 7.3 | | 0.003 | 167 | 0.07 |
| | mdamd134 | 5.1 | 4.93 | 0.0036 | 216 | 0.154 | | lncap | 0.5 | | 0.00015 | 441 | 0.12 |
| | | | | | | | | 22rve | 0.5 | | 0.0002 | 138 | 0.06 |
| **cervix** | me-180 | 224 | | 0.081 | 245 | 0.088 | **stomach** | | | | | | |
| | hela | 0.09 | 0.4 | 0.00002 | 224 | 0.065 | | hutu80 | 0 | 0 | 0 | 1 | 0.0006 |
| | | | | | | | | nugc4 | 1801 | 1314 | 0.47 | 91 | 0.03 |
| **colon** | caco2 | 99.5 | 183 | 0.032 | 137 | 0.045 | | mkn74 | 10 | 1.65 | 0.001 | 243 | 0.05 |
| | ht29 | 1105 | 917 | 0.31 | 147 | 0.05 | | mkn1 | 1.65 | 2.85 | 0.0016 | 258 | 0.14 |
| | snuc1 | 1473 | 1384 | 0.36 | 3.3 | 0.0008 | | nugc3 | 1.8 | 1.85 | 0.0009 | 251 | 0.12 |
| | skco1 | 1966 | 1479 | 0.35 | 326 | 0.077 | | ags | 91 | 145 | 0.055 | 255 | 0.1 |
| | | | | | | | | hcg27 | 0.2 | 0.5 | 0.0002 | 129 | 0.06 |
| **esoph** | fadu | 228 | 196 | 0.066 | 201 | 0.058 | | mkn45 | 560 | 546 | 0.2 | 206 | 0.075 |
| **liver** | hepg2 | 750 | 485 | 0.17 | 177 | 0.61 | | | | | | | |
| | hep3b | 3.1 | 3.1 | 0.0014 | 248 | 0.115 | | | | | | | |
| | huh7 | 5.6 | 7.5 | 0.0027 | 238 | 0.086 | | | | | | | |
| | plc/prf56 | 78.5 | 63 | 0.024 | 136 | 0.052 | | | | | | | |

Current papers using cell lines to study GPXs employ the standard 10% FBS/FCS (fetal calf serum/fetal bovine serum) formulation and do not supplement media with Se. We will reference the use of DepMap and THPA as sources of cell line expression level data on *GPX*s where generally no Se supplementation was used with 10% serum. A paper by Touat-Hamici et al. comes closest to addressing the disparities between mRNA levels, protein levels and activity for *GPX*s between the use of 10% FCS without Se (15 nM; mean level for the list of FCS/FBS vendors in [80]) and with supplementation to the near optimal level of 45 nM [79]. GPX activity was depressed two-fold by the lower Se condition in two lines expressing high levels of *GPX2*, while GPX activity was depressed four-fold in two lines with *GPX1* and no *GPX2*. The TPM values for individual cell lines can only be suggestions for the potential of *GPX* expression because of this practice. The lack of Se supplementation also means that some results may not be reproducible by others or even the same group due to variation in Se levels in different batches of serum [80]. We found that using TPM for the cell lines used in many papers *GPX2* expression levels constitute 6%–20% of total antioxidant gene TPM with one exceptional line at 47% (Table 1). The manipulation of expression levels around these values via silencing (si-RNA, sh-RNA), and overexpression produced significant effects in cell line behavior and resistance to stresses. This implies that 6% of total antioxidant enzyme gene TPM translates into abundant enough GPX2 activity to impact total antioxidant enzyme activity on ROOH. This is one reason we set our *GPX2* TPM significance threshold to 64 (3–6% total TPM). It seems probable that such claims are made for lesser expression levels involving overexpression of *GPX2* in cell lines with very low basal *GPX2* expression levels. The uncertainty has to do with incomplete data presentation and discrepancies in expression levels among different sources of information

generally for THPA/DepMap vs. published data with some among the published sources. A lack of Se supplementation reducing the impact of GPXs causes us to have a skeptical attitude for universally accepting claims using cell lines [21,77–80].

### 2.5. Building Antioxidant Enzyme Expression Profiles in Tumor-Derived Cell Lines

One more source of information to be mined for building profiles of antioxidant gene expression is cancer-derived cell lines (DepMap and THPA). The expression level metric is based on reads converted to TPM, like the reported values for tumors and normal tissues (Figures 4, 9 and 10).

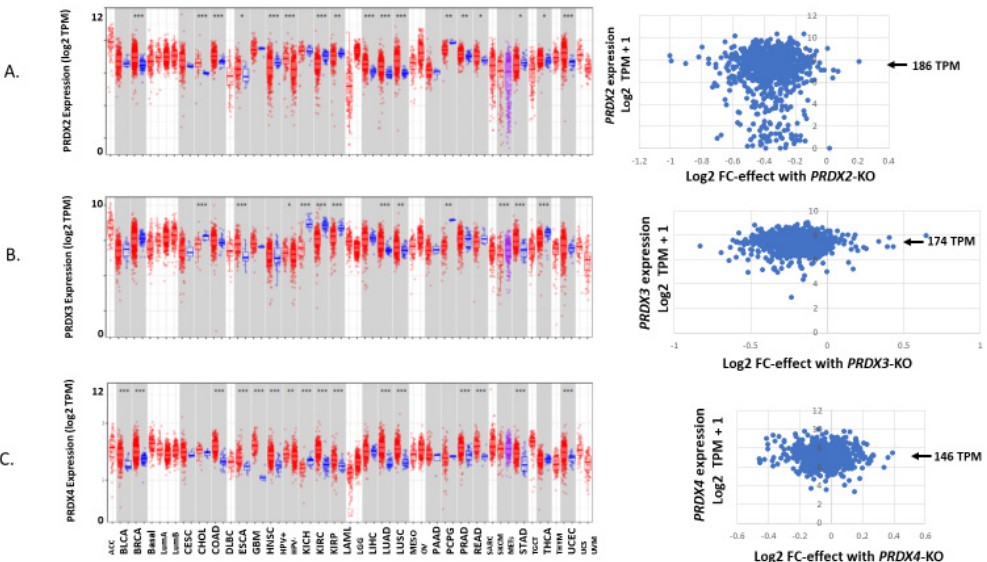

**Figure 9.** A continuation of Figure 4 showing the expression of *PRDX2* in normal and tumors paired with tumor-derived cell lines (**A**), *PRDX3* (**B**) and *PRDX4* (**C**).

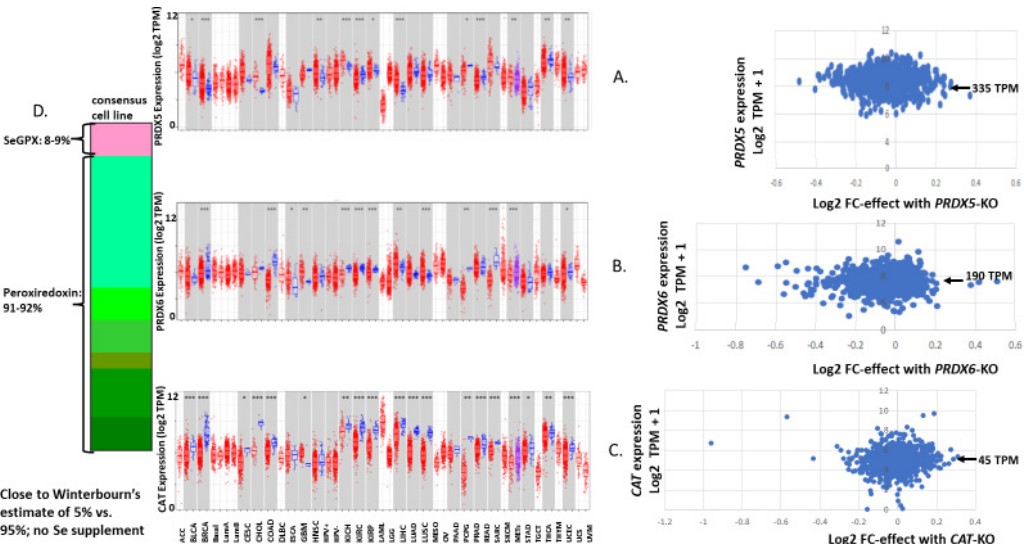

**Figure 10.** A continuation of Figure 4 showing the expression of *PRDX5* (**A**), *PRDX6* (**B**) and *CAT* (**C**). Panel (**D**) shows the antioxidant enzyme gene expression profile for a consensus tumor-derived cell line, summing up the median TPM values for each enzyme from the values shown in Figures 4, 9 and 10.

This produces a similar outcome with *GPX2* expression values ranging over 2900-fold for the collection of 1400+ cell lines (July 2023—updated semi-annually) and 2900-fold within the stomach cancer set. It is common for individual sets representing different cancer types to span 500–1500-fold (Figure 11). The range is not unique to *GPX2*. *GPX1*

and *PRDX2* have a small population of low-expressing cell lines. The general range of expression for *PRDX*s and catalase is 16–32-fold (Figures 4, 9 and 10).

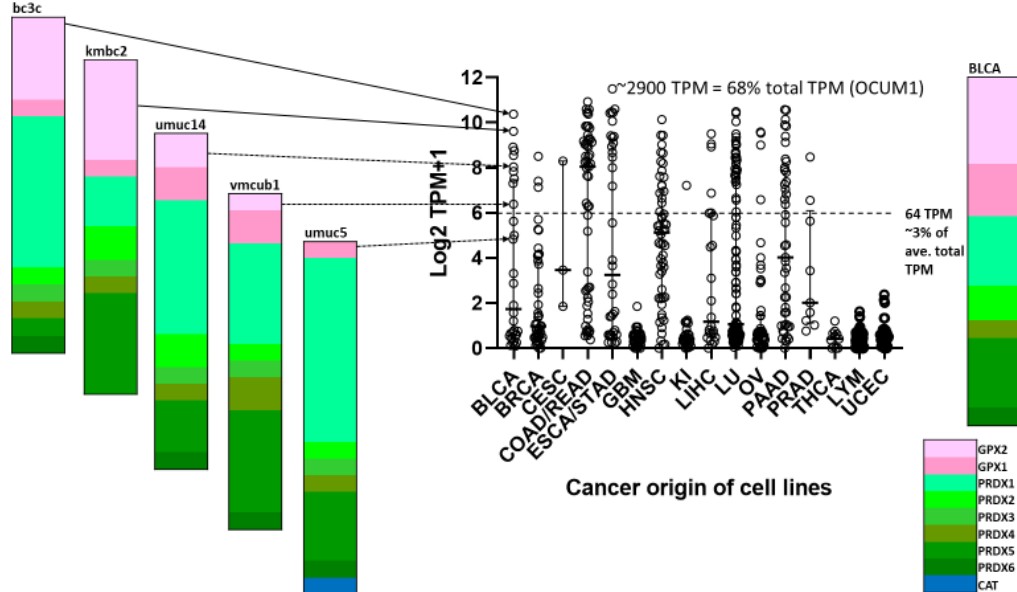

**Figure 11.** Tumor-derived cell line variation in *GPX2* expression divided up by origin of tumors. KI = kidney, LU = lung, OV = ovary; LYM includes thymus and lymphoid-derived lines. Some categories are combined in DepMap as indicated. Variation in expression is illustrated for five lines derived from BLCA. The current record holder for high *GPX2* expression is OCUM1, in terms of raw TPM and percentage of total TPM.

The data set is best regarded as a snapshot, since only one value is reported for any cell line and there is no tangible way to estimate the 95% confidence interval. We had to fill in some cell line profile data from THPA. In comparing *GPX2* TPM data sets between DepMap and THPA, we found a correlation with an R-square value of 0.94 for 41 cell lines that have been used in publications, although the residuals in the upper range tend to get quite large, providing some sense of the uncertainty in TPM data (Figure 12).

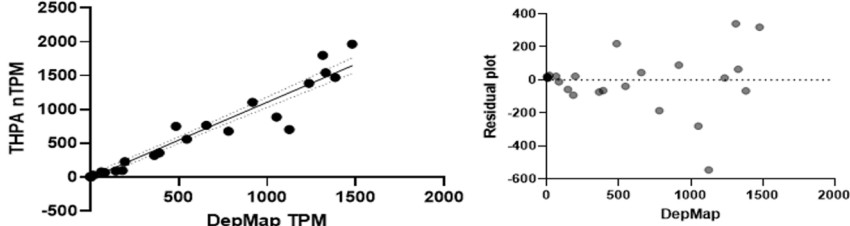

**Figure 12.** Correlation of DepMap and THPA *GPX2* TPM in tumor-derived cell lines used in publication on GPX2 (black dots and solid line) with 95% confidence interval for the trend line (dotted lines), (**left**) panel. Residuals plot, (**right**) panel.

This adds a means of obtaining more accurate TPM estimates, averaging DepMap and THPA values, with the slight problem that THPA and DepMap do not overlap for all cell lines (Table 1). DepMap has dependency scores as log2 fold-change (Log2FC), showing which genes are essential for cell growth and viability (Figures 4, 9 and 10) [10].

Within sets of cell lines, we can have some confidence that the median should be a reliable indication of the expression level for the collection. We can play with this idea to compare the outcome of compiling antioxidant gene expression levels with an estimate of relative ROOH consumption based on protein abundance for a "typical" cell and relative ROOH rate constants presented by Christine Winterbourn [54]. In Winterbourn's estimate, PRDXs would account for 95–97% of ROOH consumption, while GPXs would account

for last 3–5%. When we select the median expression value for *GPX*s, *PRDX*s and *CAT* and compile a "typical" cancer-derived cell line profile, *PRDX*s account for 91–92% of the total antioxidant enzyme gene TPM, and Se*GPX*s account for 8–9% (arrows in Figures 4, 9 and 10, indicating the median log2 TPM converted to TPM). As a first approximation, the relative TPM seems to agree with Winterbourn's estimates (Figure 10D). Both are largely based on culture media with 10% serum and no Se [81,82]. Given that *PRDX* and *CAT* expression levels would not be impacted by Se status in culture, we can build total antioxidant enzyme gene expression profiles that still show a fair number of cell lines where GPX2 could be contributing substantially to ROOH metabolism (Figure 11; up to 68% of the total TPM; OCUM1 gastric cancer cell line). We illustrate a set of examples of profiling for bladder cancer-derived lines showing the range of *GPX2* expression as a percentage of total antioxidant enzyme gene expression (Figure 11; compare to BLCA). We must remain aware that *GPX1* mRNA might be underrepresented in many cell lines due to the use of 10% serum without Se supplementation, and although *GPX2* mRNA is supposed to be resistant, we have one example where this may not be true [79].

A second evaluation of the *GPX1* and *GPX2* expression levels can be made for individual cell lines or small collections of lines by comparison with results from publications [72,83–88]. We looked at mRNA profiles and GPX activity for a modest-sized set of breast cancer-derived cell lines and a few other lines from other cancer types, most expressing only *GPX1*, a few co-expressing *GPX1* and *GPX2* and one with only *GPX2* (MDAMB175), this from our early work [85,86]. The mRNA profiles (Northern blots) from our studies and a study by Al-Taie OH et al. (HCT116, CaCo2, HCT15 and LoVo cell lines) coincided with the results gleaned from DepMap [3,83,85]. While we supplemented our culture media with Se, it may be instructive to see how much the GPX activity levels can be predicted from the DepMap data set (Figure 13).

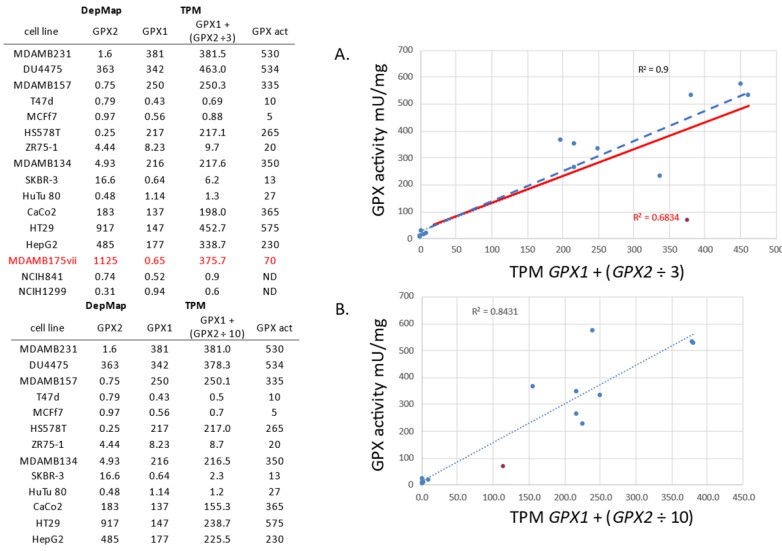

**Figure 13.** GPX activity data in cell lines from [83,85] vs. *GPX1* and *GPX2* expression data from DepMap. The activity data is from cell line culture with Se supplementation. Two simulations are shown for the combining of GPX1 and GPX2 TPM for comparison to the activity data. One follows our estimate that GPX2 has 1/3 the activity of GPX1 in the coupled assay (panel (**A**)) and the second with GPX2 having 1/10th activity (panel (**B**)). A similar fit is achieved for both if MDAMB175 is tossed from the *GPX2* ÷ 3 set (Blue line). With it included, the *GPX2* ÷ 10 set looks better. ND = not done. Red font denotes a GPX2 only expressing cell line. The red line in panel A shows the trend line omitting this GPX2 expressing only line from the correlation.

This also serves to relate DepMap TPM to GPX activity levels (or at least the potential for GPX activity) in the case where Se levels were near optimal and could represent values in patient samples (USA subjects) [83,85]. For example, levels up to ~10 TPM (ZR75-1) yield

very low GPX activity, while values for MDAMB157 (250 TPM) predict GPX activity near that of human breast samples [83,85]. The R-square value of the correlation of GPX activity to the TPM values for the lines was ~0.9, an exception made for the MDAMB175 cell line (Figure 13; observe two simulations of the data, one with *GPX2* TPM ÷3 and the second with *GPX2* TPM ÷ 10 in accordance with the two estimates of GPX2-specific activity). In DepMap, it is subline vii of MDAMB175 that is used. We were unaware of sublines for MDAMB175. We agree with DepMap that MDAMB175 can express prominent levels of *GPX2* and almost no *GPX1*. As detailed by Esworthy, Baker and Chu, there were problems growing MDAMB175 in a standard media formulation, under which GPX2 was detected as an activity, as a protein by Se-75 labeling on SDS-PAGE and as an mRNA on Northern blots. Altering the culture media based on conditions for growing another problematic cell line, DU4475, improved growth while eliminating *GPX2* expression [83]. The range of expression from DepMap for *GPX1* was about 800-fold, while the range of GPX activity in the sets was 100-fold. We had a panel of NSCLC cell line mRNA that we probed for *GPX1* and *GPX2*. In agreement with DepMap, we identified two lines (NCIH841 and NCIH1229) as having neither *GPX1* nor *GPX2* expression (Table in Figure 13) [83]. The DepMap values for the cell line, HeLa, also seem to agree with reports in the literature, having almost no *GPX2* expression and mid-level *GPX1* expression [60,88,89]. Another issue found in a recent paper was the representation of a cell line called LNCaP as having high GPX2 levels via Western blotting [90], while three other papers counter this by showing LNCaP to have very low levels of GPX2 (Western and Northerns); consulting DepMap and THPA also quickly settles this issue [4,66,79].

Four papers report on *GPX2* mRNA levels based on RT-PCR in small sets of cells lines from specific cancer types [72,84,86,87]. The cell lines are cultured in media without Se supplementation. Two papers are redeemed by using xenografts of the cell lines in mice where the resulting tumors would be Se sufficient. The results support the in vitro findings [84,87]. The general result is that in ranking the cell lines for *GPX2* expression levels, the publications and the DepMap results largely agree. The R-square values for correlations across the studies were 0.65, 0.8, 0.87 and 0.91 (Figure 14).

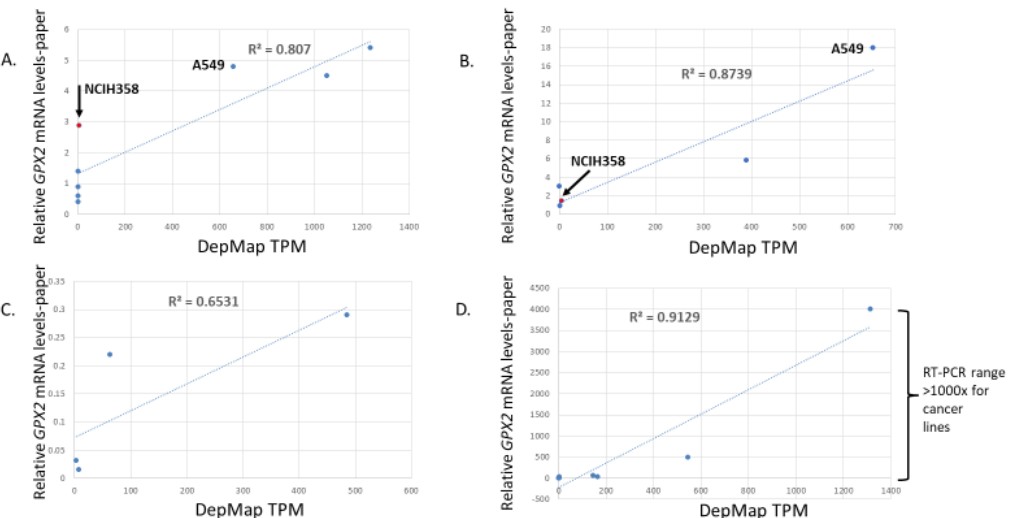

**Figure 14.** Correlation of DepMap *GPX2* TPM expression levels for cell lines used in papers and the RT-PCR results for each. Ref. [86] (panel (**A**)), 72 (panel (**B**)), 87 (panel (**C**)), 84 (panel (**D**)). Arrows and labels in panels (**A**,**B**) point out discrepancies between two papers for relative values of lines NCIH358 and A549.

However, the range of values in three publications is only a few folds versus the general 500 to 1500-fold from the DepMap website. There is one exception where stomach cancer-derived cells lines had a more similar 4000-fold range for the paper and 1300-fold in DepMap for the same lines [84]. As to what to trust, we note that two low-expressing lung

cancer-derived cell lines as evaluated via RT-PCR (NCIH1792 and SW1573) had detectable GPX2 protein on Western blots, something that would seem impossible based on the low TPM values in DepMap (0.36 and 1.7 TPM) [86]. This is important since the authors of this paper used overexpression of *GPX2* as a part of the study. The overexpression achieved in the two low-expressing lines was 8-fold based on Western blots. If we accept the combined information from RT-PCR and Western blot results presented in the paper, this should get the *GPX2* levels to the range of the high-expressing lines in the set. If we accept the DepMap results, 8-fold overexpression would do almost nothing (8 × ~1 TPM mean = 8 TPM vs. 980 TPM, mean for the three high-expressing lines). This paper does have an anomalously high relative value for the NCIH358 cell line vs. DepMap (3 vs. 5; 5 for a high-expressing cell line in common between two papers (A549), relative expression), whereas in a second paper, the relative expression levels conform a little closer to DepMap (1.5 vs. 18 for A549; 135-fold in DepMap; 4.8 TPM and 650 TPM) (Figure 14). We are left with the impression that if the goal is to find lines that span diverse levels of expression, DepMap expression data might be used to select cells for study. However, investigators might be disappointed that the span of that range is not replicated by in house RT-PCR and occasionally will find that the ranking is not quite as expected. As a precaution, we would recommend using sets of cell lines as opposed to a single model cell line.

We did come across a curious case where a mouse glioma cell line cultured as tumors in mice was evaluated for gene expression by both RT-PCR and FPKM [91]. The FPKM metric implied *GPX2* expression amounted to ~0.003% of the total antioxidant enzyme gene profile (0.75/26,000–42,000 total FPKM) across four sets of experimental conditions, with only small variation (range, 0–4 FPKM; note that a positive signal was reported for none of the samples in one set, one of three samples in two sets, and two of three samples in the last set). When evaluated via RT-PCR, two experimental conditions showed 4-fold- and 200-fold-higher levels than the control (1 FPKM) against lower FPKM expression in one set (0.66 FPKM) and a slight increase in the last (1.25 FPKM). This was not peculiar to *GPX2*. Several other genes displayed a similar bias in the same conditions when using RT-PCR versus FPKM. The low *GPX2* FPKM expression in the mouse glioma cell line is consistent with levels reported in human gliomas and derived cell lines by all the databases examined [92]. This represents one more instance of the inherent differences between the outcome of the TPM/FPKM methodology and routine RT-PCR.

In our 30 years review, we also noted problems with RT-PCR, the main point being confusion between detection of *GPX2* mRNA and significant levels of expression [21]. There are problems dealing with very low levels of expression. In one paper, we have two presentations of RT-PCR results using the same samples to demonstrate different points. In one case, the difference between the two sample sets is about 50,000-fold. In the next figure, the difference is closer to 500-fold [93]. We presume the problem derives from the low-end values which were set to the reference value of 1 in each of the figures. We know they are low since they are only 0.001% of HepG2 levels. Likewise, in a paper also mentioned in our review, the authors describe *GPX2* expression in rat liver as "trace", although the notation < 0.02 (ratio to GAPDH) is also supplied [94]. We interpret this to mean that a signal was found in at least one of the eight available samples, like the FPKM results mentioned for mouse glioma tumors. If discrepant results can occur within one study, it seems likely that different investigators will generate different outcomes from RT-PCR of the same samples as illustrated by the case of NCIH358 and A549, above.

The discussion of LNCaP, the two low *GPX2*-expressing lung cell lines and HeLa above leads to another point about methodology and data presentation. Western blots were used at some point in the papers. Initially, we thought this was a step in the right direction, showing the protein could be detected. For LNCaP, GPX2 was detected in several papers (0.69 TPM; DepMap), although in one study, Se supplementation was required before this was possible. Even with Se supplementation, the GPX2 levels were near the limit of detection, 2% of HepG2 (485 TPM; DepMap) [79]. In this latter case, we have the context of LNCaP expression levels against HepG2 and both in the context of the presence

and absence of Se. The results suggest that Western blotting can have incredible sensitivity to the presence of GPX2, and this could be abused. In the case of the lung paper, the 8-fold overexpression is shown via Western blotting [85]. We see it demonstrated against the parent cell lines and not the high-expressing lung cancer lines. Similarly, knock-down of *GPX2* is demonstrated for two high-expressing lung lines relative only to the parental lines. This robs us of the full context of how the manipulations alter expression across the full set of lines.

In the case of Hela, RT-PCR in one study showed 130,000-fold less expression than ME180, consistent with results from DepMap and THPA (Table 1) [88]. ME180 expresses high but not extraordinary levels of *GPX2* (Table 1; THPA). The following panel in the paper shows no detectable protein in HeLa by Western blotting compared to ME180. The next panel shows that the protein was detectable in HeLa in the context of comparing the impact of *GPX2* overexpression, opting not to show this in the full context of HeLa and ME180. This suggests that researchers can manipulate exposures to suit narratives and somehow the GPX2 protein is detectable even when mRNA levels are near zero (RT-PCR and DepMap). That GPX2 could be detected in Hela and LNCaP in any context, Se supplementation or not, seems amazing. We speculate that there is some sort of background issue here, although we would have no idea how this arises. Clearly, there is no dishonesty involved. This is more the fault of reviewers not insisting on seeing a more complete context of the impact of silencing or overexpression, which deprives the reader. The incongruities among different methodologies and research outcomes, TPM/FPKM vs. RT-PCR vs. IHC (ref. [92]) vs. western vs. DCFH-DA, keep accumulating [21,92].

## 2.6. Tumor Purity Metric for Adjusting Apparent Changes in GPX2 Levels

The next paper up for comment looks across tumor categories where smoking may be a factor for evidence of a relationship between *GPX2* expression levels and immune infiltration [95]. Dividing oral cavity (OCSCC; HNSC), lung (LUAD and LUSC) and bladder (BLCA) tumor sets into cold and hot (relative immune cell infiltration levels), they looked at differentially regulated genes among the sets. *GPX2*, *AKR1C1*, *AKR1C3*, *CP4F11* and *CYP4F3* were found to be up-regulated in cold tumors of all four sets, *GPX2* most notably. Up-regulation was tied to NRF2 activation and correlated with *TP63* in a panel of 65 HNSC cell lines as indicated by prior work and reproducible in DepMap [39]. The added feature of this study was the use of tumor purity estimates to standardize levels of up-regulation of *GPX2* in cold tumors, acknowledging the general lack of *GPX2* expression outside the tumor boundaries and resulting in lower estimates for the fold changes (Figures 6–8; https://gdc.cancer.gov/about-data/publications/pancan-aneuploidy (accessed on 1 July 2023); https://api.gdc.cancer.gov/data/4f277128-f793-4354-a13d-30cc7fe9f6b5 (accessed on 1 July 2023)) [64]. The results suggest that at least one-half of the increase in *GPX2* levels was derived from passive passenger effects, while one-half might represent real increases at the cellular level. The paper addresses the meaning of this residual increase using cell line studies. The value of this paper is that after adjustment for tumor purity, the perceived difference in *GPX2* levels is put on more solid ground even though the magnitude has been reduced.

## 2.7. Kaplan–Meyer Survival Curves Based on High and Low GPX2 Levels (TPM or IHC Criteria)

Kaplan–Meier survival curves are essential for cancer prognosis, primarily focusing on overall survival and recurrence-free survival. Patient samples are often categorized into high- and low-GPX2 expression groups based on either raw TPM data or IHC scoring. While both methods lack the ability to distinguish cell-to-cell variation accurately, providing intensity and proportion scores should yield more informative results. Both methods are often used without comparison to normal, a rational choice based on the range of expression levels among many cancers. The use of the combined intensity and proportion scores for dividing samples is no worse than using the TPM data as TPM data are completely blind when distinguishing cell-to-cell variation in levels of expression vs. proportion of the tumor

cell population in the samples. It falls just short of ideal over the conflation of the IHC intensity score and proportion of expressing cells. It is more informative to also evaluate intensity and proportion as lone parameters, looking for a correlation between the two and using the combination scoring. The reason for this is prognosis based on raw TPM or combined intensity and proportion scores from IHC may be stating the obvious, that more advanced cancer in terms of invasion into an organ (proportion) and opportunity for metastasis is bad (shown in ref. [72]), more so when high GPX2 levels are related to poor prognosis (the majority finding). GPX2 is merely the messenger of this sad fact (passenger effect).

One error that is made in a few papers is setting the TPM cut-off too high or low to make any real sense; this would impact studies using IHC as well. Two studies look at LUAD prognosis dividing the samples at the median value. In one paper, the prognosis results are equivocal ($p = 0.14$) [96]. In the second paper, the outcome is just at the cut-off for significance, indicating that high expression favors poor prognosis ($p = 0.046$) [97]. Both papers' cut-off of ~8 TPM probably places too many essentially non-expressing tumors in the high category, based on the negative IHC results for normal tissues presented above and normal TPM values being at most 16 TPM. The low end for LUAD tumor purity estimates is 0.2 [64]. However, we will use the 0.13 estimate here. Applying this to 8 TPM establishes an over estimation of the median value of 61 TPM for a tumor purity of 1. The LUAD cell line, EKVX, has this level of *GPX2*, amounting to 1.7% of total antioxidant enzyme gene TPM (*GPX1*-157 TPM). In using the survival analysis tool in THPA, the optimum recommended cut-off is 254 TPM (~127 FPKM). We found that ~128 TPM produced a *p* score of 0.013. This roughly divides the LUAD data set within the upper bound of the 3rd quartile (~162 TPM; 27% of set, high *GPX2*). The high range also includes many outliers that extend to 2000 TPM. Applying two estimates for median LUAD tumor purity of 0.4 and 0.6, our cut-off translates to 260–400 TPM at a value of 1. NCIH1648 falls in this range (350 TPM), showing *GPX2* levels to be 10.5% of total antioxidant gene TPM (*GPX1*-119 TPM). Based on our earlier assessment of potentially significant expression starting from a low of 6%, a cut-off of 128 raw TPM seems much more reasonable than using the median value and more likely to have prognostic value.

A third paper, already discussed, looked at both LUAD and LUSC [72]. This study used IHC criteria to divide the sample, limiting the high category to the upper third of the combined NSCLC samples. The scoring used a four-tier intensity evaluation (no staining = 0, low = 1, intermediate = 2 or high intensity = 3) multiplied by a proportion score (0 (0%), 1 (1–25%), 2 (26–50%), 3 (51–75%) and 4 (>75%). In addition to not breaking down the Kaplan–Meyer curves by intensity or proportion scores alone, they used the same high vs. low division for LUAD and LUSC. Using THPA, we can estimate that the cut-off corresponds to approximately 240 TPM, placing 34% of the NSCLC samples in the high category. We would argue that LUAD and LUSC should be assessed by different standards given the large difference in expression values, both by IHC (22 high of 158 samples, LUAD; 64 high of 94 samples, LUSC) and TPM criteria (LUAD median = 8 TPM; LUSC median = 230 TPM). The argument is also based on evidence that LUAD and LUSC originate from different cell types [73]. The combined cut-off is far too high for LUAD (see above) and possibly too low for LUSC (600 TPM is recommended at THPA). In terms of the IHC criteria in the paper, this seems to work out to a cut-off at >2 for LUAD and ≥8 for LUSC.

*2.8. Matching Cell Lines to Clinical Data Sets to Test Hypotheses about Potential Impact of GPX2 and Mechanism of Action*

Once *GPX2* TPM levels have been adjusted for tumor purity, it is possible to find cell lines that match tumor values for the high and low cut-off threshold that would be used for Kaplan–Meyer analysis. For example, the lung cancer-derived cell line A549 was used in two recent studies. In one case, the cancers examined included LUSC, which we have seen has very high *GPX2* expression levels, as does A549 [72]. In the second study, the same cell line is used in a study of LUAD [97]. As indicated, only a fraction of LUAD tumor samples

have high *GPX2* levels even accounting for low tumor purity. The use of this cell line by itself in the second study is questionable. The use of multiple cell lines with expression levels around 60–125 TPM would have enhanced the value of the paper by more closely matching the *GPX2* levels in a major proportion of the tumors. DepMap and THPA provide a means to better match cell lines to the characteristics of the tumors.

*2.9. GPX2, Tumor Immune Environment, Metabolomics and Proteomics*

More recent work has explored GPX2 in the broader context of cancer immunity and metabolic effects, leading to greater understanding of the down-stream impact of GPX2 and how alterations in expression levels are translated into effects such as epithelial–mesenchymal transition (EMT), enhanced migration, invasion, growth, and angiogenesis. Some of these studies employ more updated proteomic and metabolomic methods to advance the field.

2.9.1. Proteomics Merged with mRNA Analyses

Old-school proteomics generally used 1D (SDS-PAGE) and 2D gel electrophoresis (various 2nd dimensions) followed by staining of proteins with Coomassie brilliant blue R-250 and later silver stains. This was enhanced in the 1980s when matrix-assisted laser desorption/ionization (MALDI) and tandem mass spectrometry became available. This permitted the protein spots from 2D gels or proteins derived from other fractionation methods to be identified, usually after protease digestion (see references below).

Anecdotally, we worked next to a lab at RPMI that studied PRDX6 in the 1970s, although under the name LTW-4 (*LTW4 → AOP2 → PRDX6*) [98]. It was easily detected among liver and kidney proteins by Coomassie brilliant blue R-250 staining of 2D gels, facilitating the identification of charge variants among mouse strains. GPX1 has a more variable representation on 2D-gels depending on the source. Hammad et al. were able to identify PRDX2 (protein spot not numbered on gel) and PRDX6 (mid-abundance protein on gel; two spots) on 2D gels, but not GPX1, although this was one of their stated goals [99]. In colon cancer-derived SW480 cells grown without Se supplementation, GPX1 was barely detected until the cells were treated with CAPE [100]. In rat liver, GPX1 appears as abundant as PRDX6 (one protein spot) [101]. In mouse liver, GPX1 appeared to have 1/3rd the abundance of PRDX6 (3 spots) [102]. In human liver, relative GPX1 and PRDX6 protein levels in the range of mouse and rat liver would be predicted from the TPM levels. GPX1 was identified as a low-abundance protein in mouse stem cells on 2D gels before differentiation that declined in abundance after differentiation (Se supplementation not mentioned) [103]. Bypassing gels, freshly harvested human nasal epithelium tissue was processed for fractionation and analysis with 2D-LC-MS/MS [104]. PRDX1, 2, 3, 4, 5, 6, GPX1 and CAT were identified among the proteins in the soluble fraction, GPX2 was not (relative abundance; 1.06: 0.38: 0.08: 0.05: 0.28: 0.16: 0.07: 0.1, respectively). A second project started with 1D gels of protein derived from cultured human colon samples (undifferentiated; no Se supplementation) used to make spheroid cultures (differentiated; no Se supplementation) [105]. The study found relative values of 1 for PRDX1, 0.19 for PRDX2, 0.25 for PRDX3, 0.069 for PRDX4, 0.35 for PRDX5, 0.31 for PRDX6, 0.14 for GPX1 and 0.1 for GPX2 in the differentiated state; 1 for PRDX1, 0.23 for PRDX2, 0.3 for PRDX3, 0.13 for PRDX4, 0.23 for PRDX5, 0.22 for PRDX6, 0.14 for GPX1 and 0.03 for GPX2 in the undifferentiated state (CAT is not found). As indicated, PRDX6 is historically easy to identify, while GPX1 and presumably GPX2 seem to be below or at the border line of detection in several cases, with GPX1 approaching or reaching the level of PRDX6 in three instances. In the two cases where we have more complete information, the results seem consistent with Winterbourn's presentation of GPX and PRDX abundances and generally fit the profile of compiled TPM for our consensus cancer-derived cell line, particularly the undifferentiated colon cell cultures (Figure 10).

It is both individual and collective abundances of the six PRDXs that propelled them to notice as having a major role in the regulation of ROOH levels [106]. Later, the peroxide rate

constants of PRDXs were upwardly revised. This takes us to the view espoused in Figure 10, based on the combination of relative abundances and similarity of the rate constants to Se-GPXs [54,107]. We are using mRNA levels (TPM/FPKM) to roughly simulate the relative protein abundances.

Se-GPXs can be more readily identified and quantified using Se isotope incorporation into the proteins, a method heavily employed by us in the 1990s (radioactive 75Se in our work); those currently used are non-radioactive 76Se and 77Se [23,83,108]. The use of 75Se labeled proteins from cell lines fractionated on SDS-PAGE in combination with activity assays was used in our estimation of relative GPX1- and GPX2-specific activity, mentioned above.

Proteomic studies on GPX2 have sometimes started with 1D gel SDS-PAGE or 2D-gel electrophoresis followed by tryptic digestion and combined with MS and MS/MS or MALDI-TOF [105,109]. These methods were used to identify alteration in levels accompanying phenomena such as culture of colon organoids, as mentioned above, and to find changes in protein composition between wild-type mouse and GPX2-KO mouse colon tissues. The outcomes from the second study were integrated with results from RT-PCR.

A study on the effect of quercetin on rat colon integrated mRNA analysis using Affymetrix Gene Chip arrays and with a proteomics approach that used 2D-LC-MS/MS [110]. *GPX2* mRNA was identified as up-regulated in colon after treatment of rats with quercetin. However, the analysis failed to identify GPX2 among the significantly up-regulated proteins. In fact, the discussion suggests the likelihood it was not detected at all.

### 2.9.2. Tumor Immune Environment

We have already mentioned one paper that examined the tumor immune environment [95]. There are three others on this topic. However, we find this one to be the most complete and compelling as it merges metabolomics with mRNA data, cell line studies and xenografts to explore GPX2 involvement with the immune environment (Human Metabolome Database (V.4.0); MetaboAnalyst (V.5.0). It appears to us that GPX2 was not the initial driving force behind the study. Rather, it seems to have been discovered during the investigation. TCGA RNA-seq data were used to define the immune environment of cancers as described above. This is not what we expected as TIMER2.0 was designed for this type of analysis. Additional genes identified in this analysis were 20 chemokines/chemokine receptors, 19 interleukins and 17 human leukocyte antigens. Work using the HNSC cell lines, FaDu with high GPX2 expression (Table 1) and UMSC22A (described as low GPX2 expression, appears comparable to FaDu on Westerns; no data in DepMap or THPA), silencing of expression of *GPX2* increased production of PGE2, IL-6 and other pro-inflammatory cytokines and overexpression in the mouse MOC1 cell line lowered PGE2 production. GPX2 involvement in prostaglandin metabolism has been explored [6]. Overexpression also increased 3- hydroxykynurenine levels, which might be consistent with finding on kynureninase levels being up-regulated with *GPX2* silencing in the gastric cancer [84]. Tumors generated from MOC-1 cell in C57Bl6 mice showed that *GPX2* overexpression reduced T-cell and M1 macrophage infiltration and did not affect M2 macrophages. We point out that HNSC tumors do not have a high *GPX2* median level (80 unadjusted TPM; 5.5% total TPM) but have a range of expression from nearly zero to > 3000 TPM. Median *GPX1* levels are about three-fold higher, but do not have elevated high-end levels (~1000 TPM). Prognosis was better with low *GPX2* expression in OCSCC. This is suggested to follow from a more favorable immune environment for tumor suppression.

An in silico analysis of *GPX2* involvement with pancreatic cancer explored the immune environment [111]. The paper starts with the observation that *GPX2* is up-regulated in pancreatic adenocarcinoma (PAAD) using data from the TCGA database with increasing expression occurring from stage 1 to stage 2. PAAD does have high *GPX2* expression (313 TPM; 16% total TPM; *GPX1*-378 TPM, 20% total). This paper skips over an analysis of prognosis. THPA suggests that low expression would be favorable, using a very high

cut-off (2× median of 280 TPM) and the outcome is not significant. Using the median value shows absolutely no difference. In looking at immune cell abundances at distinct stages of the cancer, they noted that M2 macrophages levels increased from stage 1 levels as the cancer progressed. The M2 marker, *CD163*, showed a negative correlation with *GPX2*. The data for *GPX2* levels seems to be at odds with what we see at TCGA, being far too low, but the trends are found there. Also, the correlation between *GPX2* and *CD163*, while technically significant, does not look strong to us. The negative correlation with *GPX2* shows up in the other papers, with one raising the point that high M2 infiltration is associated with poor prognosis [112].

The next study is about COAD and combines analysis of *GPX2* expression with microsatellite instability (MSI), largely using an in silico analysis [113]. Using THPA at the optimum cut-off (just over the median), the authors show high expression favors good prognosis. *GPX2* is highly expressed here as already commented on. Along with STAD and READ, there is less variation in the expression range (6–32-fold) than for several other cancers (Figure 4A). This paper begins by analyzing differentially regulated genes associated with MSI, finding *GPX2* to have lower expression in the MSI positive set. Uterine cancers also show this, with the magnitude of the negative correlation with MSI more striking for COAD. Tumor mutation burden is also negatively correlated with *GPX2* expression in COAD. Several cancers share this, but most of them are sites of very low *GPX2* expression, kidney, prostate and thyroid. Looking at immune cell infiltration, they found a negative correlation with M2 macrophages and several categories of immune cells. In this case, high *GPX2* levels were interpreted as suppressing mutations that might fuel more aggressive tumors. They also linked the high expression to fewer M2 macrophages, which should favor a good prognosis.

Finally, Yang et al. looked at the immune cell composition of low and high-expressing prostate tumors. While the results appear equivocal to us, the authors suggest that *GPX2* levels affected the relative infiltration of eight classes of immune cells, which includes a negative correlation with M2 macrophages, consistent with the papers presented above [90].

### 2.9.3. Metabolomics and GPX2

This first paper is more about determining how the potential anti-cancer agent, $\alpha$-hederin, acts on NSCLC and its ability to sensitize cancer cells to ferroptosis after cis-platin treatment [114]. A combination of proteomics, metabolomics and high-throughput sequencing suggested the involvement of GPX2 and glutathione synthase down-regulation in the increased susceptibility to ferroptosis, this using A549 line and PC9 (not found in DepMap; Western blot analysis suggests slightly lower GPX2 levels in PC9 vs. A549). The use of cell lines was limited to control vs. $\alpha$-hederin-treated. Additional support for ferroptosis was found in the proteomics study and using inhibitors selective for death-related pathways, Z-VAD (caspase inhibitor), chloroquine (autophagy inhibitor), necrostatin-1 (necroptosis inhibitor) and ferrostatin-1 (ferroptosis inhibitor).

The second paper looked at *GPX2* in gastric cancer [84]. The initial approach is standard for this type of work, mixing analysis from the databases with overexpression or silencing of *GPX2* in cell lines. The cell line studies looked at proliferation, migration and EMT as end points. Higher *GPX2* levels enhanced proliferation, migration and EMT. Proteomic analysis linked low *GPX2* expression to suppressed kynurenine metabolism by up-regulating kynureninase. This appeared to be triggered by allowing the accumulation of ROOH. The link to proliferation, migration and EMT is proposed to be through the aryl hydrocarbon receptor and its ligand kynurenine. Given the general indication that *GPX2* expression is high in normal and cancerous stomach tissues, this is a plausible hypothesis (526 TPM; median level, 30% of total TPM). The paper acknowledged the co-expression of *GPX1* in cell lines and, in cancer and normal samples, found no difference in levels. Their in-house Kaplan–Meyer analysis (IHC) and the experimental components of the study seemed to favor low expression of GPX2 for better prognosis. This was attributed to lower kynurenine levels negatively impacting migration and invasion potential of cells.

The cell lines used were NUGC4, NUGC3 and MKN45 with 1314 TPM (47% total TPM), 1.85 TPM (0.09% total TPM) and 546 TPM (20% of total TPM), respectively. RT-PCR and Western blot analysis agreed well with the DepMap values. RNA silencing was used on NUGC4 and MKN45 (~5-fold and 10-fold lower levels, respectively, via Western blots) while overexpression was used for NUGC3. The use of NUGC3 is the one minor failing of the paper. We never obtain the true context for how much the overexpression elevates the GPX2 protein level or whether the mRNA levels reach our cut-off of 6% total TPM. The two higher expressing lines have TPM values that would seem to place them in the range of the tumor samples with the simplifying assumptions that that $\frac{1}{2}$ of the tumor specimens have tumor purity values > 0.6 and the highest levels of expression probably involve purity values close to 1 (Figure 4A). The only real downside is the lack of Se supplementation for the in vitro work, which might yield a disconnect between the TPM levels and the protein levels. This is partially redeemed in the use of xenografts from the NUGC4 and MKN45 lines that are more likely to be Se sufficient, like the host, and their use in this manner replicated the in vitro findings.

Ren et al. found a link between GPX2 and angiogenesis in tumors, largely based on results from a mouse tumor model [115]. Ingenuity pathway analysis showed low GPX2 levels were associated with ROS/hypoxia-inducible factor-$\alpha$ (HIF1$\alpha$)/vesicular endothelial growth factor A (VEGFA) signaling, causing poor perfusion and hypoxia via defective angiogenesis. We discussed this paper in our recent review. Considering the current discussion, we note that the paper is based on the observations that *GPX2* levels tend to decline in breast cancer. Our main criticism would be lack of information on the levels of *GPX2* in the mouse tumor cell lines before and after manipulation of *GPX2* expression levels in relation to levels in human tumors and cell lines. This could have been performed as the authors presented a sub-study using two human breast cancer-derived lines, MDAMB231 and MDAMB361. MDAMB231 has *GPX2* expression like that of Hela and LNCaP cells, while MDAMB361 has moderate levels of *GPX2* representing 6% of total TPM (Table 1). Thus, MDAMB361 falls into our range of potentially significant GPX2 expression. We lack context on the impact of *GPX2* overexpression on MDAMB231. However, going by the effect of *GPX2* silencing in MDAMB361 on growth of xenografts and the replication of the effects on HIF1$\alpha$ and VEGFA levels, the mouse tumor results are confirmed in the study of the human cell line. Exactly how the human cell line *GPX2* levels relate to tumor progression is still uncertain. The only breast cancer cell line that might have GPX2 levels like those constructed from the results of Kannan et al. for breast basal/stem cells is MDAMB175vii (Table 1) [21]. All the rest of the available lines have much lower levels of *GPX2* agreeing with the suggestion from the public databases that *GPX2* levels decline from normal to tumor. However, the bulk of breast cancers originate from luminal cells, where normal *GPX2* levels would be low at the outset [70].

## 3. Limitations

One limitation of this project is the admitted confusion of the commentators over incongruent results found by comparing results from papers in fine detail, both among themselves and with public database information. At the midpoint of our investigation, we thought the problems lie with the TPM/FPKM metric used in the databases and the snapshot nature of the data collection. At the end, we are convinced the discrepancies lie more with the methodology of published studies.

Our approach is one of convenience. Since the available data are of mRNA levels, which is what we are forced to use. IHC results are too few and too inconsistent for use in this project. We have tried to use TPM/FPKM data to approximate relative and total antioxidant enzyme levels in tissues and cell lines. Obviously, this is a big leap with the potential for huge error. The outcomes do roughly agree with the scant proteomic data [106,107]. We are simplifying the portrayal of rate constant similarity among Se-GPXs, PRDXs and CAT. These have tended to change as the years progress, not only in the big leap proposed by Winterbourn for PRDXs but in fine detail later [54,107,116]. Then, there is

the issue of PRDX inactivation via ROOH, which remains an open question that cannot be addressed by these methods [117]. Se-GPX may not be immune to this [118].

We have pointed out some of the problems involved in building antioxidant enzyme gene profiles of normal and tumor tissues. There may be more problems with this approach than we are aware of at this time. In terms of making a judgment about significant levels of *GPX2*, we were forced to rely on the very information we are critiquing, cell line data in reference to DepMap, and THPA TPM data. Our threshold of 6% seems low to us and is tainted by the general lack of Se supplementation in the culture media of most studies. However, we cannot outright dismiss the outcomes of so many papers. We can call into question results based on lower percentages. There must be some limit for a significant impact of GPX2. In our 30 years review, we pointed out that a positive outcome was achieved using the DCHF-DA assay by silencing *GPX2* in Hela cells. The authors admitted they could not detect GPX2 in the parent line via Western blotting [119]. We expect readers of this commentary to be as skeptical of our approach as we are to claims of GPX2 effects in HeLa cells.

## 4. Summary

In writing this commentary, we are trying to provide investigators, editors, reviewers and readers of past, current and future papers on this topic with perspective and mention tools that can be employed to evaluate the strengths and possible weakness in the research and by researchers to design better conceived studies.

First, we strongly recommend the use of Se in cell culture, a practice lost in recent work. There are two points here: one, vendors have the Se levels for each batch of serum; two, sodium selenite is not expensive and readily available. Along these lines, we propose that a big-science project be undertaken jointly by members of the selenium/redox fields to assay GPX activity, perform Western blots and measure *GPX* TPM for cell lines representing *GPX1* expression alone, *GPX2* expression alone (at least 5 lines; DepMap) and a mix of the two, extending the work of the authors, Brigelius-Flohe and co-workers, Touat-Hamici et al. and others mentioned in this commentary [77,79,83,85]. This could be combined with some test of cell line resistance to oxidants after silencing of each isoenzyme along the lines of Shen and Nathan to determine if there is a lower limit of GPX expression that can be rated as significant and how dependent this is on total antioxidant enzyme expression levels [40]. Such a project could be joined to DepMap. The project should be justifiable as legitimate cancer research both from the number of claims for GPX1-4 involvement in cancer and Se impact on human health, including cancer.

Second, work presenting the impact of silencing or overexpression should be presented in the full context of all lines used in work, possibly using reference lines such as MDAMB231 and MDAMB361, ME180 and HeLa, HepG2 and Huh7, and ME180 and HeLa (Table 1). This would extend to work using both RT-PCR and Western blotting. This could be relegated to the supplemental data if the availability is mentioned in the main text. The data presentation model of Touat-Hamici et al. is recommended [79].

Third, using DepMap and THPA, as we have done here, would remind investigators of the presence and possible great abundance of *GPX1* and *PRDX*s in the cell lines they choose for the study of *GPX2*. We would hope that claims for *GPX2* might be tempered by parallel manipulation of other antioxidant enzyme genes (selecting one more based on expression levels in DepMap). This would not diminish the claims for GPX2. In our view, it would strengthen the hypothesis that GPX2 is operating via modulation of ROOH and, coupled with the proper assays, might provide an estimate of how much ROOH GPX2 is handling. Only two recent papers even bother to present data on *GPX1*, with one paper showing a similar impact on the stresses of hydrogen peroxide and LPS after silencing in the Het-1A esophageal epithelial cell line [34,84].

Fourth, the use of tumor and cell line databases serves to evaluate whether *GPX2* levels are high enough to have any impact in a particular cancer type and whether alterations in levels contribute to the view that *GPX2* levels might stray into the range of significant

expression and be worth study. Studies linking *GPX2* to glioma, prostate, and breast cancer (bulk of LUAD samples) should be subject to greater scrutiny given that *GPX2* levels might be too low to have any impact based on the analyses presented here [90,93,117]. One of our major complaints is that papers with suspect links between *GPX2* and cancer are cited as support for *GPX2* involvement in cancer [84,115,120].

Fifth, cell lines can be better matched to tumor sample expression levels to make any conclusions derived from manipulation of *GPX2* more meaningful. This would be of some greater concern for the tissues mentioned in item four. We have demonstrated a few instances of how this could be accomplished.

**Author Contributions:** R.S.E. conceived the topic of the commentary and wrote the first draft. F.-F.C. initiated the use of public databases in our studies, reviewed the draft of the paper and made edits to the draft. All authors have read and agreed to the published version of the manuscript.

**Funding:** This research received no external funding.

**Data Availability Statement:** All data in this article is derived from published sources or the indicated public databases. Further information can be sought by emailing the authors.

**Acknowledgments:** The results shown here are in part based upon data generated by the TCGA Research Network: FPKM and cancer purity data; The Human Protein Atlas (IHC images, Kaplan–Meyer overall survival data and some cell line TPM data); TIMER2.0 (Normal and cancer specimen TPM (log2 TPM plots); and Cancer Dependency Map Project (tumor-derived cell line TPM data and effect data); data and images downloaded between September 2022 and July 2023.

**Conflicts of Interest:** The authors declare no conflict of interest.

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
