# Peer review of "Using Information from Public Databases to Critically Evaluate Studies Linking the Antioxidant Enzyme Selenium-Dependent Glutathione Peroxidase 2 (GPX2) to Cancer"

_biomedinformatics, doi:10.3390/biomedinformatics3040060_

Round 1

Reviewer 1 Report

Comments and Suggestions for Authors

The major goal of this manuscript is to provide helpful insights to the scientific community involved in GPX2 research. It underlines the need of using effective tools for assessing study strengths and addressing potential flaws, which can lead to better-conceived studies in this subject. The authors recommend the following techniques to accomplish this. (i) Reintroduce selenium (Se) into cell culture. (ii) For informed cell line selection, use DepMap and THPA. (iii) Examine the levels of GPX2 in tumor and cell line databases. (iv) Examine any references to GPX2-cancer connections, and (v) Match the expression levels of cell lines to tumor samples. This study is informative and interesting to the readers. Here are my comments:

1.      When assessing gene expression data, it is critical to keep tumor purity in mind. Explain how tumor purity data from the TCGA can affect the interpretation of antioxidant enzyme expression levels, particularly in tumor vs. non-tumor samples. Discuss how using raw median TPM values may impact how GPX2's contribution to total antioxidant enzyme TPM is portrayed.

2.      Integrate gene expression data with other omics data (e.g., proteomics, metabolomics) to create an overall view of the molecular landscape in cold and hot tumors could aid in discovering other variables that lead to GPX2 level variations.

3.      In line 514 please provide potential methods for increasing the accuracy and importance of GPX2-based prognostic outcomes. For example, to improve statistical power, consider experimenting with alternative cut-off values or increasing the sample size.

4.      Please avoid speculative claims regarding the possible outcomes of higher sample sizes or different approaches. Instead the authors can focus on examining the present studies' limitations and consequences.

5.      In the summary please provide additional information and explanations for each recommendation to emphasize its significance and advantages to researchers and the scientific community.

6.      Page 10, please organize the content into sections or subsections to address various aspects of the research findings. Divide the topic, for example, into sections pertaining to various cancer types or evaluation approaches.

7.      When discussing differences in GPX2 expression levels between studies or databases, explain potential reasons for these differences, such as different cell types or methodology utilized.

8.  Page 10 and 11, Some of the sentences are overgeneralizing or drawing conclusions that are not completely supported by the evidence. Please give specific proof and context for each statement.

Comments on the Quality of English Language

To eliminate ambiguity and make your views more obvious, please make your message more specific.

Please keep the language straightforward and concise to enhance readability and understanding. For example line 587-592 may be written as “Kaplan-Meier survival curves are essential for cancer prognosis, primarily focusing on overall survival and recurrence-free survival. Patient samples are often categorized into high and low GPX2 expression groups based on either raw TPM data or IHC scoring. While both methods lack the ability to distinguish cell-to-cell variation accurately, combining intensity and proportion scores yields more informative results.”

Author Response

Response to Reviewer 1.

Thank you for your patience and interest in this topic. We have tired to address all of the suggestions, leading to extensive rewrites and additional sections added to the original document.

  1. When assessing gene expression data, it is critical to keep tumor purity in mind. Explain how tumor purity data from the TCGA can affect the interpretation of antioxidant enzyme expression levels, particularly in tumor vs. non-tumor samples. Discuss how using raw median TPM values may impact how GPX2's contribution to total antioxidant enzyme TPM is portrayed.

We emphasized this point in several sections of the new draft, both as general issue and in one specific example of colon (lines 294-310; 319-321; other sections). Also, we address this as a problem of defining the source of tumors for colon, lung and breast tissues (lines 294-310; 383-391; 393-407; 415-420).

This is also addressed in a separate section -2.6

  1. Integrate gene expression data with other omics data (e.g., proteomics, metabolomics) to create an overall view of the molecular landscape in cold and hot tumors could aid in discovering other variables that lead to GPX2 level variations.

We had to place this at the end to prevent too much disruption to the flow of the main thesis (section 2.9 and subsections therein). We tried to break this down into discussions of the tumor immune environment, proteomics and metabolomics.  However, the topics do tend to spill over. The proteomics section may be too much. We got side-tracked by the issue of where Winterbourn got the relative PRDX and GPX protein abundances presented in the 2008 paper (Winterbourn CC, Hampton MB. Thiol chemistry and specificity in redox signaling. Free Radic Biol Med. 2008 Sep 1;45(5):549-61. doi: 10.1016/j.freeradbiomed.2008.05.004. Epub 2008 May 16. PMID: 18544350). There is no reference provided for the source of the data. Back in the day, we thought we knew the source of the information and agreed with the values. Trying to find anything tangible at the present time was a real task that involved a deep dive into supporting data files with only 2 real hits. Fortunately, they seem to agree with conventional views and seem to support the idea that TPM has some predictive value for estimating relative protein abundances.

  1. In line 514 please provide potential methods for increasing the accuracy and importance of GPX2-based prognostic outcomes. For example, to improve statistical power, consider experimenting with alternative cut-off values or increasing the sample size

For this, we retained the original section and added to it specific means to improve use of IHC or TPM criteria.  This includes using the THPA prognosis tool to find recommended cut-offs and adjusting them to suit various goals. We also emphasize the need to keep the cut-off points related to the likely expression levels of specific tumor types and how to establish these (main point of commentary).

  1. Please avoid speculative claims regarding the possible outcomes of higher sample sizes or different approaches. Instead the authors can focus on examining the present studies' limitations and consequences.

I think you are referring to our comparing two lung studies, commenting on the similarity of the Kaplan-Meyer curves and the different P scores.  We revised that by side-stepping a comparison in favor of generally criticizing the joint choice of median value for categorizing high expression and low expression.

  1. In the summary please provide additional information and explanations for each recommendation to emphasize its significance and advantages to researchers and the scientific community.

We beefed up the summary by adding constructive comments on how the work might be improved. We hope that this is adequate.

  1. Page 10, please organize the content into sections or subsections to address various aspects of the research findings. Divide the topic, for example, into sections pertaining to various cancer types or evaluation approaches.

We broke this up into 3 sub-sections.  The divisions are a bit arbitrary.  We think they make sense.

  1. When discussing differences in GPX2 expression levels between studies or databases, explain potential reasons for these differences, such as different cell types or methodology utilized.

This ended up being scattered over various sections of the paper as there were many comparisons to be made. We open with the issues of disparities for normal tissue values (lines 216-224). The main points are covered in lines 689-732. Here we get into the meat of the issues regarding disparities between the database information and publications. We ended up changing our minds on this as mentioned in the limitations section, lines 1024-1029. At first, we were convinced there was something wrong with the databases and on lines 216-224 we do point out some peculiarities with samples in the databases. However, on going back over the commentary for the purpose of revising it, we ended up thinking the problems are in the published studies relating to formulaic data presentation and general misunderstanding of how GPX2 would act in cells and tissues.  This is seen in how investigators handle RT-PCR, western blots and over expression/silencing data (lines-639-732). It starts slowly and builds as gets deeper into the critique.  The biggest single problem is lack of context for expression levels, both just what they mean in any cells and how to understand the context of over-expression or silencing in relation to the full range of expression in a particular set of cells or tissues. We bring this issue up at several points including the summary.

  1. Page 10 and 11, Some of the sentences are overgeneralizing or drawing conclusions that are not completely supported by the evidence. Please give specific proof and context for each statement.

We revised the section to better explain our viewpoint, which we admit may appear a bit cynical, and allow for a more balanced view of the issues involved.  The main problem here is multiple cell types in tissues giving rise to cancers.  This still presents issues major issues in understanding breast cancer in relationship to GPX2 expression-basal cells vs. basal cell cancer-not necessarily related case by case.  For us, figuring out prostate cancer seems difficult, as well. This partially derives from the inconsistent results about which cells are expressing GPX2-dueling IHC results-THPA vs paper. We didn’t find anything specific to the source cells of LUAD and LUSC in relation to GPX2 expression. For colon, there is more tangible evidence for what is going on.

Comments on the Quality of English Language

To eliminate ambiguity and make your views more obvious, please make your message more specific.

Please keep the language straightforward and concise to enhance readability and understanding. For example line 587-592 may be written as “Kaplan-Meier survival curves are essential for cancer prognosis, primarily focusing on overall survival and recurrence-free survival. Patient samples are often categorized into high and low GPX2 expression groups based on either raw TPM data or IHC scoring. While both methods lack the ability to distinguish cell-to-cell variation accurately, combining intensity and proportion scores yields more informative results.”

We went over the entire paper to get the language clearer. 

Reviewer 2 Report

Comments and Suggestions for Authors

The present manuscript weigh in on these recent findings and discuss the impact on the relative GPX2 and GPX1 functions. The present manuscript was good in overall. The flow was structured. Also, the findings provided was valuable to scientific community. I just only suggest the authors for present article's limitations should be added before moving on to the conclusion section.

Comments on the Quality of English Language

-

Author Response

We added a limitations section as requested.